# Logged tropical forests have amplified and diverse ecosystem energetics

Yadvinder Malhi[1 ✉], Terhi Riutta[1,2], Oliver R. Wearn[3], Nicolas J. Deere[4], Simon L. Mitchell[4], Henry Bernard[5], Noreen Majalap[6], Reuben Nilus[6], Zoe G. Davies[4], Robert M. Ewers[7] & Matthew J. Struebig[4]

Old-growth tropical forests are widely recognized as being immensely important for their biodiversity and high biomass[1]. Conversely, logged tropical forests are usually characterized as degraded ecosystems[2]. However, whether logging results in a degradation in ecosystem functions is less clear: shifts in the strength and resilience of key ecosystem processes in large suites of species have rarely been assessed in an ecologically integrated and quantitative framework. Here we adopt an ecosystem energetics lens to gain new insight into the impacts of tropical forest disturbance on a key integrative aspect of ecological function: food pathways and community structure of birds and mammals. We focus on a gradient spanning old-growth and logged forests and oil palm plantations in Borneo. In logged forest there is a 2.5-fold increase in total resource consumption by both birds and mammals compared to that in old-growth forests, probably driven by greater resource accessibility and vegetation palatability. Most principal energetic pathways maintain high species diversity and redundancy, implying maintained resilience. Conversion of logged forest into oil palm plantation results in the collapse of most energetic pathways. Far from being degraded ecosystems, even heavily logged forests can be vibrant and diverse ecosystems with enhanced levels of ecological function.

Human-modified forests, such as selectively logged forests, are often characterized as degraded ecosystems because of their altered structure and low biomass. The concept of ecosystem degradation can be a double-edged sword. It rightly draws attention to the conservation value of old-growth systems and the importance of ecosystem restoration. However, it can also suggest that human-modified ecosystems are of low ecological value and therefore, in some cases, suitable for conversion to agriculture (such as oil palm plantations) and other land uses[3–5].

Selectively logged and other forms of structurally altered forests are becoming the prevailing vegetation cover in much of the tropical forest biome[2]. Such disturbance frequently leads to a decline in old-growth specialist species[1], and also in non-specialist species in some contexts[6–8]. However, species-focused biodiversity metrics are only one measure of ecosystem vitality and functionality, and rarely consider the collective role that suites of species play in maintaining ecological functions[9].

An alternative approach is to focus on the energetics of key taxonomic groups, and the number and relative dominance of species contributing to each energetic pathway. Energetic approaches to examining ecosystem structure and function have a long history in ecosystem ecology[10]. Virtually all ecosystems are powered by a cascade of captured sunlight through an array of autotroph tissues and into hierarchical assemblages of herbivores, carnivores and detritivores. Energetic approaches shine light on the relative significance of energy flows among key taxa and provide insight into the processes that shape biodiversity and ecosystem function. The common currency of energy enables diverse guilds and taxa to be compared in a unified and physically meaningful manner: dominant energetic pathways can be identified, and the resilience of each pathway to the loss of individual species can be assessed. Quantitative links can then be made between animal communities and the plant-based ecosystem productivity on which they depend. The magnitude of energetic pathways in particular animal groups can often be indicators of key associated ecosystem processes, such as nutrient cycling, seed dispersal and pollination, or trophic factors such as intensity of predation pressure or availability of resource supply, all unified under the common metric of energy flux[11,12].

Energetics approaches have rarely been applied in biodiverse tropical ecosystems because of the range of observations they require[11–13]. Such analyses rely on: population density estimates for a very large number of species; understanding of the diet and feeding behaviour of the species; and reliable estimation of net primary productivity (NPP). Here we take advantage of uniquely rich datasets to apply an energetics lens to examine and quantify aspects of the ecological function and vitality of habitats in Sabah, Malaysia, that comprise old-growth forests, logged forest and oil palm plantation (Fig. 1 and Extended Data Fig. 1). Our approach is to calculate the short-term equilibrium production or consumption rates of food energy by specific species, guilds or taxonomic groups. We focus on three taxonomic groups (plants, birds

[1]Environmental Change Institute, School of Geography and the Environment, University of Oxford, Oxford, UK. [2]Department of Geography, University of Exeter, Exeter, UK. [3]Fauna & Flora International, Vietnam Programme, Hanoi, Vietnam. [4]Durrell Institute of Conservation and Ecology (DICE), School of Anthropology and Conservation, University of Kent, Canterbury, UK. [5]Institute for Tropical Biology and Conservation, Universiti Malaysia Sabah, Kota Kinabalu, Malaysia. [6]Forest Research Centre, Sabah Forestry Department, Sandakan, Malaysia. [7]Georgina Mace Centre, Department of Life Sciences, Imperial College London, Ascot, UK. ✉e-mail: yadvinder.malhi@ouce.ox.ac.uk

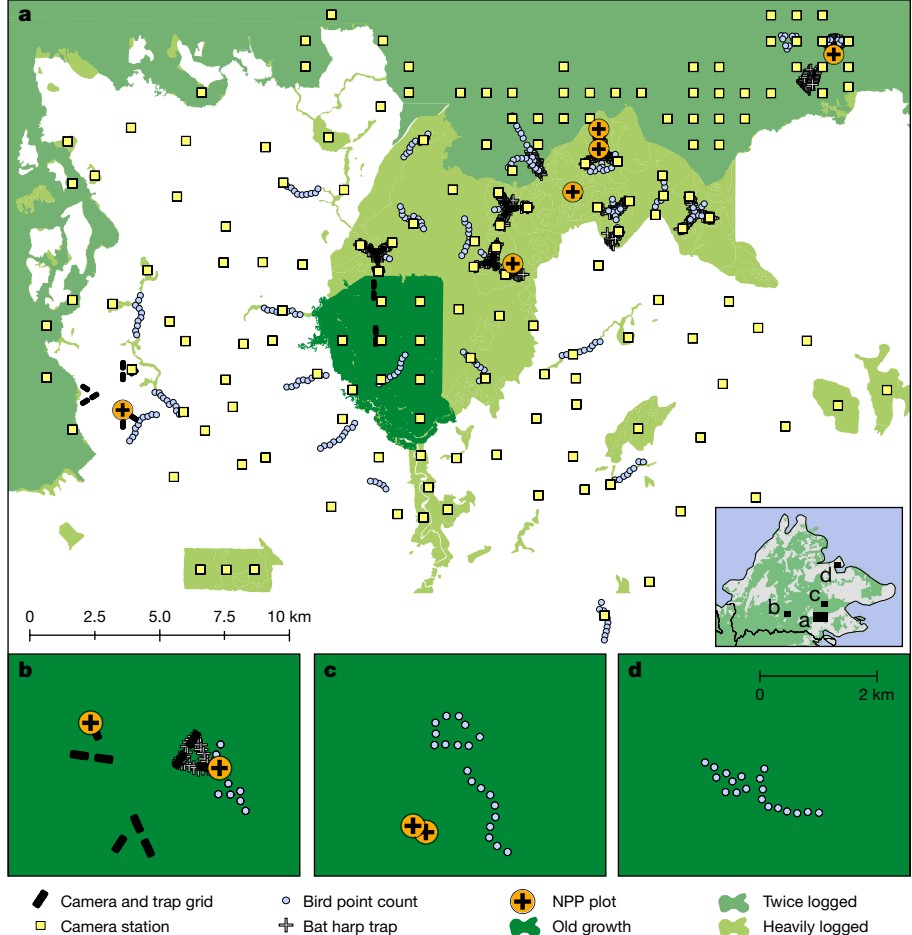

**Fig. 1 | Maps of the study sites in Sabah, Borneo. a–d**, Maps showing locations of NPP plots and biodiversity surveys in old-growth forest, logged forest and oil palm plantations in the Stability of Altered Forest Ecosystems Project landscape (**a**), Maliau Basin (**b**), Danum Valley (**c**) and Sepilok (**d**). The inset in **a** shows the location of the four sites in Sabah. The shade of green indicates old-growth (dark green), twice-logged (intermediate green) or heavily logged (light green) forests. The camera and trap grid includes cameras and small mammal traps. White areas indicate oil palm plantations.

and mammals) that are frequently used indicators of biodiversity and are relatively well understood ecologically.

We are interested in the fraction of primary productivity consumed by birds and mammals, and how it varies along the disturbance gradient, and how and why various food energetic pathways in mammals and birds, and the diversity of species contributing to those pathways, vary along the disturbance gradient. To estimate the density of 104 mammal and 144 bird species in each of the three habitat types, we aggregated data from 882 camera sampling locations (a total of 42,877 camera trap nights), 508 bird point count locations, 1,488 small terrestrial mammal trap locations (34,058 live-trap nights) and 336 bat trap locations (Fig. 1 and Extended Data Fig. 1). We then calculated daily energetic expenditure for each species based on their body mass, assigned each species to a dietary group and calculated total food consumption in energy units. For primary productivity, we relied on 34 plot-years (summation of plots multiplied by the number of years each plot is monitored) of measurements of the key components of NPP (canopy litterfall, woody growth, fine root production) using the protocols of the Global Ecosystem Monitoring Network[14–16] across old-growth (*n* = 4), logged (*n* = 5) and oil palm (*n* = 1) plots. This dataset encompasses more than 14,000 measurements of litterfall, 20,000 tree diameter measurements and 2,700 fine root samples.

Overall bird species diversity is maintained across the disturbance gradient and peaks in the logged forest; for mammals, there is also a slight increase in the logged forest, followed by rapid decline in the oil palm (Fig. 2b,c). Strikingly, both bird and mammal biomass increases substantially (144% and 231%, respectively) in the logged forest compared to the old-growth forest, with mammals contributing about 75% of total (bird plus mammal) biomass in both habitat types (Fig. 2b,c).

The total flow of energy through consumption is amplified across all energetic pathways by a factor of 2.5 (2.2–3.0; all ranges reported are 95% confidence intervals) in logged forest relative to old-growth forest. In all three habitat types, total energy intake by birds is much greater than by mammals (Fig. 2d,e and Extended Data Table 1). Birds account for 67%, 68% and 90% of the total direct consumption by birds and mammals combined in old-growth forests, logged forests and oil palm, respectively. Although mammal biomass is higher than bird biomass in the old-growth and logged forests, the metabolism per unit mass is much higher in birds because of their small body size; hence, in terms of the energetics and consumption rates, the bird community dominates. The total energy intake by birds alone increases by a factor of 2.6 (2.1–3.2) in the logged forest relative to old-growth forest. This is mainly driven by a 2.5-fold (1.7–2.8) increase in foliage-gleaning insectivory (the dominant energetic pathway), and most other feeding guilds also show an even larger increase (Figs. 2d and 3). However, total bird energy intake in the oil palm drops back to levels similar to those in the old-growth forest, with a collapse in multiple guilds. For mammals, there is a similar 2.4-fold (1.9–3.2) increase in total consumption

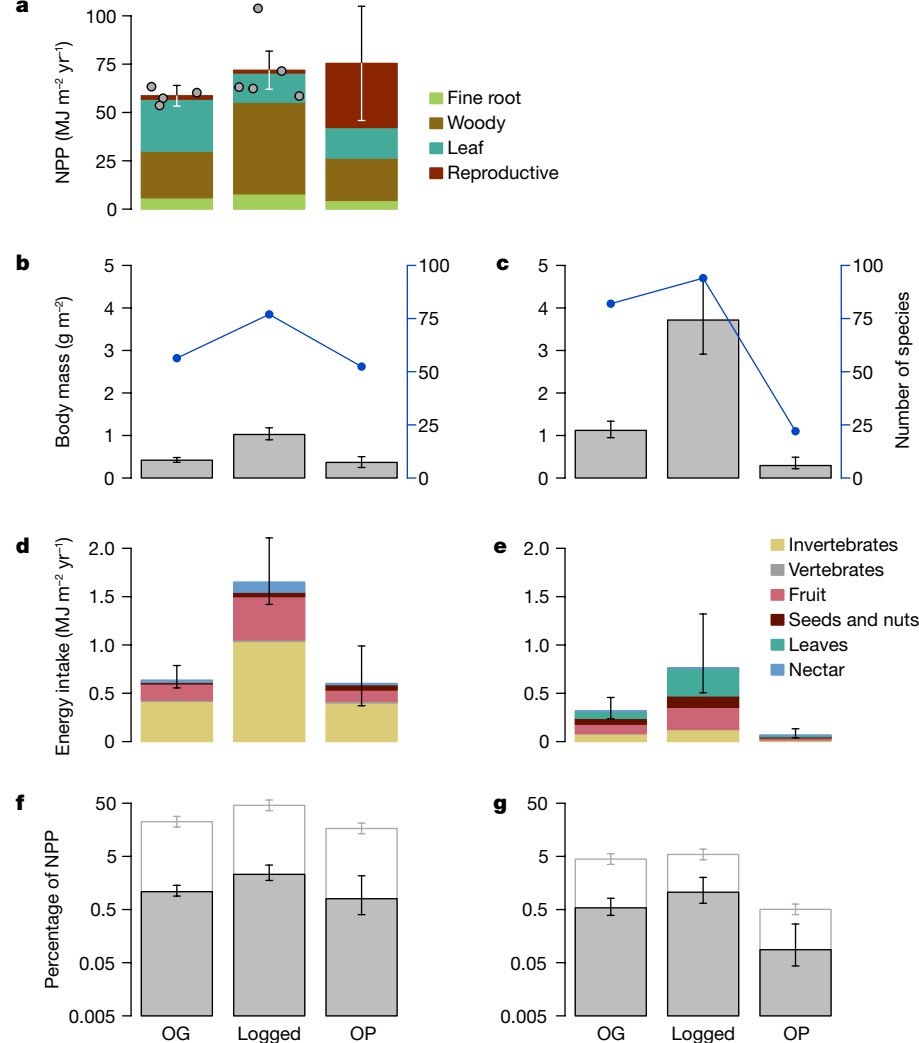

**Fig. 2 | Variation of ecosystem energetics along the disturbance gradient from old-growth forest through logged forest to oil palm. a**, Total NPP along the gradient (mean of intensive 1-ha plots; $n = 4$ for old growth (OG), $n = 5$ for logged and $n = 1$ for oil palm (OP); error bars are 95% confidence intervals derived from propagated uncertainty in the individually measured NPP components), with individual plot data points overlaid. **b**,**c**, Total body mass (bars, left axis) and number of species counted (blue dots and line, right axis) of birds (**b**) and mammals (**c**). **d**,**e**, Total direct energetic food intake by birds (**d**) and mammals (**e**). **f**,**g**, Percentage of NPP directly consumed by birds (**f**) and mammals (**g**). In **b**–**e**, body mass and energetics were estimated for individual

bird and mammal species, with the bars showing the sum. Error bars denote 95% confidence intervals derived from 10,000 Monte Carlo simulation estimates incorporating uncertainty in body mass, population density, the daily energy expenditure equation, assimilation efficiency of the different food types, composition of the diet of each species and NPP. In **f**,**g**, the grey bars indicate direct consumption of NPP, white bars denote the percentage of NPP indirectly supporting bird and mammal food intake when the mean trophic level of consumed invertebrates is assumed to be 2.5, with the error bars denoting assumed mean trophic levels of 2.4 and 2.6. Note the log scale of the $y$ axis in **f**,**g**. Numbers for **d**,**e** provided in Supplementary Data Tables 1, 2.

when going from old-growth to logged forest, but this declines sharply in oil palm plantation. Most notable is the 5.7-fold (3.2–10.2) increase in the importance of terrestrial mammal herbivores in the logged relative to old-growth forests. All four individual old-growth forest sites show consistently lower bird and mammal energetics than the logged forests (Extended Data Fig. 5).

The fraction of NPP flowing through the bird and mammal communities increases by a factor of 2.1 (1.5–3.0) in logged forest relative to old-growth forest. There is very little increase in NPP in logged relative to old-growth forests (Fig. 2a) because increased NPP in patches of relatively intact logged forest is offset by very low productivity in more structurally degraded areas such as former logging platforms[14],[15]. In oil palm plantations, oil palm fruits account for a large proportion of NPP, although a large fraction of these is harvested and removed from the ecosystem[17]. As a proportion of NPP, 1.62% (1.35–2.13%) is directly consumed by birds and mammals in the old-growth forest; this rises to

3.36% (2.57–5.07%) in the logged forest but drops to 0.89% (0.57–1.44%) in oil palm (Fig. 2f,g and Extended Data Table 2).

If all invertebrates consumed are herbivores or detritivores (that is, at a trophic level of 2.0), and trophic efficiency is 10% (ref. [10]), the total amount of NPP supporting the combined bird and mammal food intake would be 9%, 16% and 5% for old-growth forest, logged forest and oil palm, respectively. However, if the mean trophic level of consumed invertebrates is 2.5 (that is, a mix of herbivores and predators), the corresponding proportions would be 27%, 51% and 17% (Fig. 2f,g). As insectivory is the dominant feeding mode for the avian community, these numbers are dominated by bird diets. For birds in the old-growth forests, 0.35% of NPP supports direct herbivory and frugivory, but around 22% of NPP (assumed invertebrate trophic level 2.5) is indirectly required to support insectivory. The equivalent numbers for birds in logged forest are 0.83% and 46%. Hence, birds account for a much larger indirect consumption of NPP. Bird diet studies in old-growth and

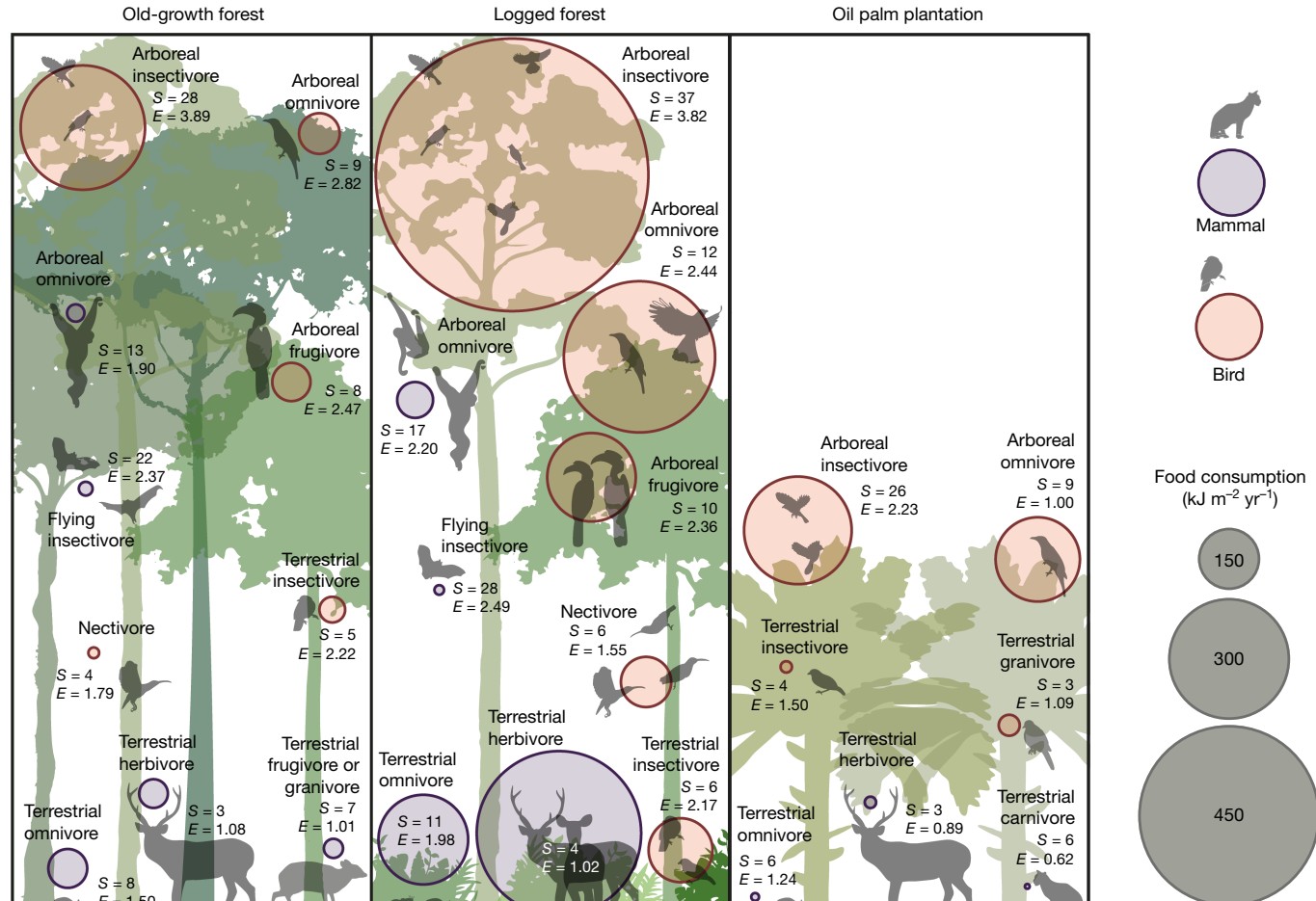

**Fig. 3 | Magnitude and species diversity of energetic pathways in old-growth forest, logged forest and oil palm.** The size of the circles indicates the magnitude of energy flow, and the colour indicates birds or mammals. *S*, number of species; *E*, ESWI, an index of species redundancy and, therefore, resilience (high values indicate high redundancy; see main text). For clarity, guilds with small energetic flows are not shown, but are listed in Supplementary Data 4. Images created by J. Bentley.

logged forest in the region suggest that consumed invertebrates have a mean trophic level of 2.5 (ref. [18]; K. Sam, personal communication), indicating that the higher-end estimates of indirect NPP consumption (that is, around 50% in logged forests) are plausible.

It is interesting to compare such high fractions of NPP to direct estimates of invertebrate herbivory. Scans of tree leaf litter from these forests suggest that just 7.0% of tree canopy leaf area (1–3% of total NPP) is removed by tree leaf herbivory[14,16], but such estimates do not include other pathways available to invertebrates, including herbivory of the understorey, aboveground and belowground sap-sucking, leaf-mining, fruit- and wood-feeding, and canopy, litter and ground-layer detritivory. An increase in invertebrate biomass and herbivory in logged forest compared to old-growth forest has previously been reported in fogging studies in this landscape[19]. Such high levels of consumption of NPP by invertebrates could have implications on ecosystem vegetation biomass production, suggesting, first, that invertebrate herbivory has a substantial influence on recovery from logging and, second, that insectivorous bird densities may exert substantial indirect controls on ecosystem recovery.

The distributions of energy flows among feeding guilds are remarkably stable among habitat types (Fig. 3), indicating that the amplified energy flows in the logged forests do not distort the overall trophic structure of vertebrate communities. Overall bird diet energetics are dominated by insectivory, which accounts for a strikingly invariant 66%, 63% and 66% of bird energetic consumption in old-growth forest,

logged forest and oil palm, respectively. Foliage-gleaning dominates as a mode of invertebrate consumption in all three habitat types, with frugivory being the second most energetically important feeding mode (26%, 27% and 19%, respectively). Mammal diet is more evenly distributed across feeding guilds, but frugivory (31%, 30%, 30%) and folivory (24%, 38%, 26%) dominate. Small mammal insectivores are probably under-sampled (see Methods) so the contribution of mammal insectivory may be slightly greater than that estimated here. The apparent constancy of relative magnitude of feeding pathways across the intact and disturbed ecosystems is noteworthy and not sensitive to plausible shifts in feeding behaviour between habitat types (see Supplementary Discussion). There is no evidence of a substantial shift in dominant feeding guild: the principal feeding pathways present in the old-growth forest are maintained in the logged forest.

When examining change at species level in the logged forests, the largest absolute increases in bird food consumption were in arboreal insectivores and omnivores (Fig. 4a and Extended Data Fig. 2a). In particular, this change was characterized by large increases in the abundance of bulbul species (*Pycnonotus* spp.). No bird species showed a significant or substantial reduction in overall energy consumption. In the oil palm plantation, total food consumption by birds was less than in logged forests, but similar to that in old-growth forests. However, this was driven by very high abundance of a handful of species, notably a single arboreal omnivore (yellow-vented bulbul *Pycnonotus goiavier*) and three arboreal insectivores (*Mixornis bornensis*, *Rhipidura javanica*,

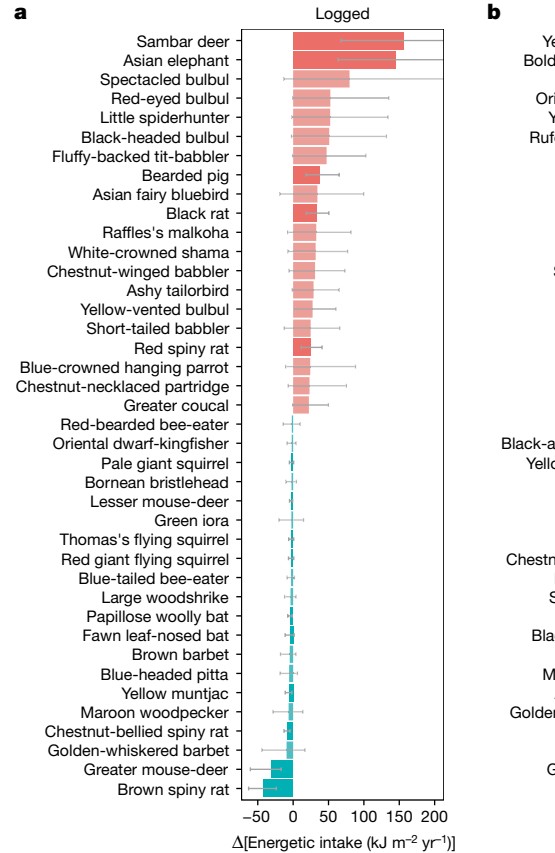

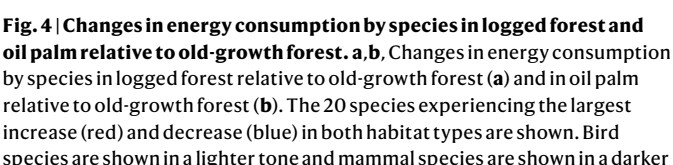

**Fig. 4 | Changes in energy consumption by species in logged forest and oil palm relative to old-growth forest. a,b,** Changes in energy consumption by species in logged forest relative to old-growth forest (**a**) and in oil palm relative to old-growth forest (**b**). The 20 species experiencing the largest increase (red) and decrease (blue) in both habitat types are shown. Bird species are shown in a lighter tone and mammal species are shown in a darker tone. The error bars denote 95% confidence intervals, derived from 10,000 Monte Carlo simulation estimates incorporating uncertainty in body mass, population density, the daily energy expenditure equation, assimilation efficiency of the different food types and composition of the diet of each species.

*Copsychus saularis*), whereas energy flows through most other bird species were greatly reduced (Fig. 4b and Extended Data Fig. 2b).

For mammals, the increase in consumption in logged forests is dominated by consumption by large terrestrial herbivores increasing by a factor of 5.7 (3.2–10.2), particularly sambar deer (*Rusa unicolor*) and Asian elephant (*Elephas maximus*; Fig. 4a and Extended Data Figs. 2b and 3), along with that by small omnivores, predominantly rodents (native spiny rats, non-native black rat; Fig. 4). A few rainforest species show a strong decline (for example, greater mouse-deer *Tragulus napu* and brown spiny rat *Maxomys rajah*). In the oil palm, most mammal species collapse (Fig. 4b) and the limited consumption is dominated by a few disturbance-tolerant habitat generalists (for example, red muntjac *Muntiacus muntjak*, black rat *Rattus rattus*, civets), albeit these species are at lower densities than observed in old-growth forest (Extended Data Fig. 2).

With very few exceptions, the amplified energy flows in logged forest seem to retain the same level of resilience as in old-growth forest. The diversity and dominance of species within any pathway can be a measure of the resilience of that pathway to loss of species. We assessed energetic dominance within individual pathways by defining an energetic Shannon–Wiener index (ESWI) to examine distribution of energy flow across species; low ESWI indicates a pathway with high dependence on a few species and hence potential vulnerability (Fig. 3). The overall ESWI across guilds does not differ between the old-growth and logged forest ($t_{2,34} = -0.363$, $P = 0.930$), but does decline substantially from old-growth forest to oil palm ($t_{2,34} = -3.826$, $P = 0.0015$), and from logged forest to oil palm ($t_{2,34} = -3.639$, $P = 0.0025$; linear mixed-effects models, with habitat type as fixed effect and guild as random effect; for model coefficients see Supplementary Table 3).

Hence, for birds, the diversity of species contributing to dominant energetic pathways is maintained in the transition from old-growth to logged forests but declines substantially in oil palm. Mammals generally show lower diversity and ESWI than birds, but six out of ten feeding guilds maintain or increase ESWI in logged forest relative to the old-growth forests but collapse in oil palm (Fig. 3). Terrestrial herbivory is the largest mammal pathway in the logged forest but is dependent on only four species and is probably the most vulnerable of the larger pathways: a few large mammals (especially sambar deer) play a dominant terrestrial herbivory role in the logged forest. In parallel, bearded pigs (*Sus barbatus*), the only wild suid in Borneo, form an important and functionally unique component of the terrestrial omnivory pathway. These larger animals are particularly sensitive to anthropogenic pressures such as hunting, or associated pathogenic pressures as evidenced by the recent precipitous decline of the bearded pig in Sabah due to an outbreak of Asian swine fever (after our data were collected)[20].

Vertebrate populations across the tropics are particularly sensitive to hunting pressure[21]. Our study site has little hunting, but as a sensitivity analysis we explored the energetic consequences of 50% reduction in population density of those species potentially affected by targeted and/or indiscriminate hunting (Extended Data Fig. 4). Targeted hunted species include commercially valuable birds, and gun-hunted mammals

(bearded pig, ungulates, banteng and mammals with medicinal value). Indiscriminately hunted species include birds and mammals likely to be trapped with nets and snares. Hunting in the logged forests lowers both bird and mammal energy flows but still leaves them at levels higher than in faunally intact old-growth forests. Such hunting brings bird energetics levels close to (but still above) those of old-growth forests. For mammals, however, even intensively hunted logged forests seem to maintain higher energetic flows than the old-growth forests. Hence, only very heavy hunting is likely to 'offset' the amplified energetics in the logged forest.

The amplified energetic pathways in our logged forest probably arise as a result of bottom-up trophic factors including increased resource supply, palatability and accessibility. The more open forest structure in logged forest results in more vegetation being near ground level[22,23] and hence more accessible to large generalist mammal herbivores, which show the most striking increase of the mammal guilds. The increased prioritization by plants of competition for light and therefore rapid vegetation growth strategies in logged forests results in higher leaf nutrient content and reduced leaf chemical defences against herbivory[24,25], along with higher fruiting and flowering rates[19] and greater clumping in resource supply[9]. This increased resource availability and palatability probably supports high invertebrate and vertebrate herbivore densities[25]. The act of disturbance displaces the ecosystem from a conservative chemically defended state to a more dynamic state with amplified energy and nutrient flow, but not to an extent that causes heavy disruption in animal community composition. Top-down trophic factors might also play a role in amplifying the energy flows in intermediate trophic levels, through mechanisms such as increased protection of ground-dwelling or nesting mammals and birds from aerial predators in the dense vegetation ground layer. This might partially explain the increased abundance of rodents, but there is little evidence of trophic release at this site because of the persisting high density of mammal carnivores[26]. Overall, the larger number of bottom-up mechanisms and surge in invertebrate consumption suggest that increased resource supply and palatability largely explains the amplification of consumption pathways in the logged forest. An alternative possibility is that the amplified vertebrate energetics do not indicate amplified overall animal energetics but rather a large diversion of energy from unmeasured invertebrate predation pathways (for example, parasitoids); this seems unlikely but warrants further exploration.

Oil palm plantations show a large decline in the proportion of NPP consumed by mammals and birds compared to logged forests[12]. Mammal populations collapse because they are more vulnerable and avoid humans, and there is no suite of mammal generalists that can step in[27,28]. Birds show a more modest decline, to levels similar to those observed in old-growth forests, as there is a broad suite of generalist species that are able to adapt to and exploit the habitat types across the disturbance gradient, and because their small size and mobility render them less sensitive to human activity[29]. There is a consistent decline in the oil palm in ESWI for birds and especially for mammals, indicating a substantial increase in ecosystem vulnerability in many pathways.

In conclusion, our analysis demonstrates the tremendously dynamic and ecologically vibrant nature of the studied logged forests, even heavily and repeatedly logged forests such as those found across Borneo. It is likely that the patterns, mechanisms and basic ecological energetics we describe are general to most tropical forests; amplification of multiple ecosystem processes after logging has also been reported for logged forests in Kenya[9], but similar detailed analyses are needed for a range of tropical forests to elucidate the importance of biogeographic, climatic or other factors. We stress that our findings do not diminish the importance of protecting structurally intact old-growth forests, but rather question the meaning of degradation by shining a new light on the ecological value of logged and other structurally 'degraded' forests, reinforcing their significance to the conservation agenda[30]. We have shown that a wide diversity of species not only persist but thrive in the logged forest environment. Moreover, such ecological vibrancy probably enhances the prospects for ecosystem structural recovery. In terms of faunal intactness, our study landscape is close to a best-case scenario because hunting pressures were low. If logged forests can be protected from heavy defaunation, our analysis demonstrates that they can be vibrant ecosystems, providing many key ecosystem functions at levels much higher than in old-growth forests. Conservation of logged forest landscapes has an essential role to play in the in the protection of global biodiversity and biosphere function.

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

## Methods

### Field sites

Data from logged forests were collected across the Stability of Altered Forest Ecosystems (SAFE) Project landscape (4° 43′ N, 117° 35′ E) in Sabah, Malaysia[31], a lowland mosaic landscape of logged forest and oil palm plantation (Fig. 1). The logged forest had been through one round of selective logging (removing 113 m³ ha⁻¹) in the 1970s and an additional cumulative volume of 37–66 m³ ha⁻¹ during the subsequent rotations up until the early 2000s, which is similar to the mean extracted volume of 152 m³ ha⁻¹ within a larger, 220,000 ha area in Sabah[32,33]. The most heavily logged plots have been logged more than four times, whereas the less logged plots have been logged only twice; hence, logging intensity was high throughout the landscape. Data for oil palm plantations were collected from adjoining oil palm estates. Data for the old-growth forests (Fig. 1) were collected from the Brantian-Tantulit Virgin Jungle Reserve (a large fragment adjoining the logged forest landscape), and also in three other old-growth forest reserves in Sabah: the Maliau Basin Conservation Area (vegetation, birds and mammals), the Danum Valley Conservation Area (vegetation and birds) and Sepilok Forest Reserve (birds alone). Data collection took place between 2010 and 2017. The sample sites spanned the gradient of logging intensity and biomass observed across the landscape (Extended Data Fig. 1). The study sites have experienced very low hunting pressure compared to other areas of Borneo owing to difficult access from nearby towns and cultural factors, including the relatively limited forest use among local populations[27]. Data on every species surveyed or estimated are given in Supplementary Data 1.

### Vegetation and NPP surveys

NPP was measured in five logged 1-ha plots in the SAFE Project area with varying intensity of logging (5 years of data), in four old-growth forest 1-ha plots in the Maliau Basin Conservation Area (two plots, 4 years of data) and Danum Valley Conservation Area (two plots, 2 years of data)[14,16], and one 0.36-ha mature oil palm plot (2 years of data), following the standardized protocols of the Global Ecosystems Monitoring network[15] (Fig. 1, Extended Data Fig. 1 and Supplementary Data 3). We quantified the following NPP components: woody NPP (stems, coarse roots and branches), canopy NPP (leaves, twigs and reproductive parts) and fine root NPP[10]. All plots had at least two tree censuses for quantifying stem and coarse root NPP. Canopy NPP (litter traps) and fine root NPP (root ingrowth cores) were monitored monthly and quarterly, respectively. Oil palm plantation NPP estimates were based on palm censuses and allometry with height, monthly counts of flower bunches, fruit bunches and attached and pruned fronds combined with a one-off survey of their mass, and quarterly harvest of the root ingrowth cores.

### Mammal surveys

To characterize the terrestrial medium and large non-volant mammal community, we obtained detection/non-detection data from remotely operated digital camera traps (Reconyx HC500) between May 2011 and December 2017[27,34]. Camera traps were deployed at 882 locations, stratified across old-growth forest ($n = 236$), logged forest ($n = 539$) and oil palm ($n = 107$). Two survey designs were adopted. The first had a hierarchical, clustered design whereby cameras were placed 23–232 m apart in grids (42,877 camera trap nights, with cameras deployed on 49 consecutive nights on average[35]). The second had a systematic design with pairs of cameras spaced more broadly over the landscape at stations >1 km apart (11,403 camera trap nights, with cameras deployed on 47 consecutive nights on average[34]). In both cases, cameras were deployed 20–50 cm off the ground, disturbance to vegetation was kept to a minimum, and no baits or lures were used.

Terrestrial small mammals were surveyed between May 2011 and July 2014 using locally made steel-mesh traps, deployed at 1,488 locations stratified across the habitat types (432, 768 and 288 in old-growth forest, logged forest and oil palm, respectively[35]). Trap locations were clustered into 1.75-ha trapping grids of 12 × 4 locations with 23-m spacing. Each location was sampled using two traps (spaced 5–20 m apart) placed at or near ground level (0–1.5 m) and baited with oil palm fruit. Traps were checked for seven consecutive mornings, and captured individuals were marked using a subcutaneous passive inductive transponder tag before being released at the capture location[27,35]. Some trapping grids were sampled more than once (14 of 31 grids), and the total sampling effort was 34,058 trap nights.

For volant mammals, we used bat capture data from harp traps set in forests between April 2011 and June 2012[33]. Bats were captured at 42 sampling points in each of 12 sites (3 old-growth forests, 9 logged forests), in traps set 50–150 m apart. Up to seven traps were set across forest trails and logging skids each night and then moved to a new position the following day. Bats were marked with unique forearm bands or wing biopsies before release so that recaptured individuals could be identified and removed from analyses. No comparable data were available for oil palm as harp traps are ineffective in open habitats.

### Bird surveys

Avian point counts were conducted across 356 locations spanning forests and surrounding oil palm estates, with sites separated by 180–220 m (ref. [36]). Each count involved a single experienced observer (S.L.M.) recording all species seen and heard within an unlimited distance over a 15-min period, including birds flying over. Four counts were conducted at each site between 05:00 and 11:00 on mornings without rain between 2014 and 2016. Sites were sampled at mean intervals of 72 days between first and last visits. Three species of swift (*Aerodramus maximus*, *A. salangana* and *A. fuciphagus*) that cannot be reliably separated in most field conditions were collectively considered as *Aerodramus* spp.

### Density estimation

For the terrestrial medium and large mammals, we estimated density at each camera trap point using the random encounter model (REM)[37,38]. This approach uses information about the size of the camera trap detection zone, and the movement speeds of animals, to correct the trapping rate data (number of animal passes per unit time) and estimate density. Specifically, the parameters required for REM include, for each species: the activity level (that is, proportion of 24-h diel cycle spent active and available for detection); movement speed when active; effective detection angle of camera traps; effective detection distance of camera traps; and the trapping rate. Activity levels were estimated on the basis of the timestamps of the camera trap detections[39], and movement speeds and the detection zone parameters were estimated on the basis of animal location data recovered from the camera trap image sequences. This was possible because we 'calibrated' both camera trap locations (using an object of known size, a 1-m pole) and the specific camera trap model that we used (by taking pictures of objects of known size at known distances from the camera). This allowed us to recover the distance and angle of animals in image sequences and thereby estimate animal speed when active[40]. The effective detection angle and distance were estimated using an adapted distance sampling approach[39]. We implemented the REM using multi-species Bayesian approaches, in which species are treated as random effects and estimates for rare species, with only sparse data available, become possible by 'borrowing' information from the more common species[41]. Separate multi-species models (with land-use type included as a covariate) for activity levels, speeds and the detection zone parameters were used to estimate the posterior distributions for each species in each land use. These posterior distributions were then combined with the trapping rate data to estimate density, with bootstrapping of the data providing the uncertainty estimates (Supplementary Data 2). The final density estimates are broadly comparable with published estimates for other sites in the region.

To estimate terrestrial small mammal densities[42], we used spatially explicit capture–recapture modelling[43]. This modelling framework explicitly accounts for the fact that some individuals with home ranges at the edge of a trapping grid may not always be available for capture. The spatially explicit capture–recapture modelling approach therefore controls for variation in the effective sampling area of a trapping grid that might occur (for example, across the disturbance gradient). Separate models for each land-use type were fitted in the R package secr[44] using default parameters (that is, a Poisson distribution of animal home-range centres and a half-normal detection function) and no covariates. A buffer of 100 m around the trap locations defined the region of model integration. Sufficient data were available to estimate density for 14 species of small mammal in old-growth and logged forest. There were too few captures in oil palm to allow for model fitting.

Unlike those for the terrestrial mammals, bat data were not acquired through a repeated survey design. Therefore, densities were derived on the basis of a 20-m detection radius (that is, 0.126 ha) around each trap, and estimates were calculated as the total counts of each species per cumulative detection area in each habitat type.

We estimated mean local abundance of birds as a function of per capita detection using the Royle–Nichols model[45]. Before analysis, species-specific detection histories were constructed by pooling detection and non-detection data into discrete sampling occasions according to site visit. Our modelling framework described abundance and detection using categorical habitat-specific intercepts (old-growth forest, logged forest and oil palm), incorporating species-specific slopes and intercepts, drawn as random effects from a common community-level distribution. Model specification and checking procedures followed established protocols[34]. We scaled modelled bird abundance (number of individuals within the effective sampling area: 7,854 m² buffer around each point count) to density per square kilometre post hoc using a conversion factor of 0.785.

### Expert judgment and independent estimates

For some other mammal species that could not be reliably sampled by camera traps or small animal traps—for example, owing to obligately arboreal habitat use (some primate and squirrel species) or migratory behaviour (for example, Asian elephant *E. maximus*)—we relied on estimates based on encounter rates with these species during the course of fieldwork, or on independent studies in the same study area (Supplementary Data 1), for example, for Bornean orangutan (*Pongo pygmaeus*)[46]. Asian elephant densities in logged forest and oil palm were estimated on the basis of the observed behaviour of the single 15-elephant herd in the SAFE landscape, and for old-growth forest as an average of the low densities reported in Maliau and the higher densities in other Sabah old-growth forests[47]. Owing to bias introduced by the life histories of highly mobile birds, modelled densities of five species of hornbill (*Anthracoceros malayanus*, *Anorrhinus galeritus*, *Buceros rhinoceros*, *Rhinoplax vigil*, *Rhyticeros undulatus*) as well as great argus (*Argusianus argus*) and crested serpent-eagle (*Spilornis cheela*) were corrected using available information from the literature. Home-range estimates of each hornbill species in each habitat type were centred around the mean value and scaled to one-unit standard deviation. This was multiplied by a conversion factor of 465.3 ha based on the mean home-range reported across the seven species (radio telemetry studies; Supplementary Data 1) to calculate scaled home-range estimates for each species. Per hectare density estimates were inferred as the inverse of scaled home range. These large bird species contributed a very small part to total ecosystem energetics and hence our overall results are very insensitive to these assumptions.

### Aggregation to habitat type

As we combined data across taxa for which we needed the largest sampling effort and 'best' description of the community possible, we aggregated species abundance estimates to a single value per habitat type.

For the REM modelling, data for a given habitat were used in the model to estimate a single value of each of the required REM parameters (for example, speed, detection angle/distance, activity level, trap rate and density) in each habitat. Hence, our unit of replication is guild, which has no spatial component within habitat type and hence no spatial autocorrelation variable that can be explored.

### Daily energetic expenditure and food uptake

Daily energetic expenditure for each individual species was calculated from body mass using published multi-species allometric equations for field metabolic rates for mammals and birds[48] (see Supplementary Table 1 for the equations and parameter values). The fractions of diet deriving from each food type were assigned to each species on the basis of specialist expert judgment by three coauthors (O.R.W., N.J.D. and S.L.M.), and food uptake rates were calculated on the basis of assimilation efficiency for each feeding guild and food type[49,50] (summarized in Supplementary Table 2).

### EWSI

We define and use ESWI to examine the partition of energy flow across species by analogy to the Shannon–Wiener index of diversity that is a measure of how population abundance is spread across species

$$\text{ESWI} = -\sum_{i=1}^{n} e_i \ln(e_i)$$

in which $e_i$ is the proportion of energy flow through species $i$ in a total community of $n$ species, relative to total energy flow through the community.

### Comparison across guilds and habitat types

To test for a significant difference in ESWI across habitat types, we applied linear mixed-effects models with ESWI as a response variable, habitat type (old-growth forest, logged forest, oil palm) as a fixed effect and guild ($n = 18$, see Supplementary Data 1) as a random effect (random intercept), with variance structure to allow for unequal variances among habitat types (varIdent function, which implements a different variance for each habitat; model with variance structure is better than without, Akaike information criterion of 120.5 versus 128.3, respectively), using the nlme package[51]. Pairwise post hoc comparison of the habitats, with Tukey adjustment, was carried out using the emmeans package[52].

### Uncertainty calculation

We assumed that there was uncertainty in the following variables: body mass of species, population density, daily energy expenditure (DEE) equation, assimilation efficiency of the different food types, composition of the diet of each species and NPP. For body mass, we drew from a truncated normal distribution (lower bound = 1 g), in which the mean was the observed body mass and standard deviation was 15%. We based this standard deviation for birds on a study of tropical birds[53] and applied the same 15% for mammals for consistency, in the absence of other data[53]. For population density, we used the 10,000 bootstrapped estimates of the population density models. In addition, for birds and bats (the population density estimates of which were based on a detection radius around the sampling point), we incorporated the uncertainty in the radius by drawing from a truncated normal distribution with standard deviation of 20%, and lower and upper bounds of 50% and 150% of the estimated radius. We assigned 30% uncertainty for each of the few 'expert guess' species, which had a very minor influence in the final results. For DEE, we estimated the 95% confidence intervals for the predictions as described in ref. [48]. For assimilation efficiency, we drew from a random beta distribution, using the mean and standard deviation by food type and guild from the literature (Supplementary Table 2). For fractional diet composition, we generated a symmetrical

beta distribution, with the peak uncertainty of 20% when the food group made up 50% of a species' diet and no uncertainty when the food group made up 0% or 100% of the diet. It is possible that logging and conversion to oil palm results in systematic shifts in diet composition towards arthropods. Therefore, we also carried out two additional analyses in which the fractional consumption of arthropods (in one analysis) and leaves (in the other analysis) was increased by 30% for species that had a mixed diet. Uncertainty in NPP was drawn from a truncated normal distribution with the mean and standard deviation derived from the field data[14,16] and lower and upper bounds of the distribution set at mean ± 2 standard deviations.

To quantify the uncertainty in our estimates for energetic intake and proportion of NPP consumed, we ran 10,000 simulations, replacing the values in our original calculations with values drawn from the random distributions. First, we estimated the total uncertainty by assuming uncertainty in all components simultaneously and calculated the 2.5% and 97.5% percentiles of the simulations to derive 95% confidence intervals for our estimates. Second, to quantify how much each variable contributed to the total uncertainty, we ran sets of 10,000 simulations in which only one variable at the time had uncertainty while others were kept constant. We calculated the 5% to 95% percentile range for the uncertainty-in-one-variable-at-the-time estimates and the uncertainty-in-all-variables estimates, and the contribution of each variable to the total uncertainty considered to be the ratio of the two[54].

Uncertainty estimates of absolute consumption are dominated by uncertainty in the population density and DEE allometry of the dominant consumers (Extended Data Fig. 6). Uncertainties in diet fraction allocation, assimilation efficiency or the consumption allometric equation make relatively modest contributions. Hence, further reduction in uncertainty is best targeted in improving estimates of the abundance and producing better DEE allometries specific to the few dominant consumers (Extended Data Fig. 7), as well as better assessing key under-sampled groups such as small forest-floor and arboreal mammals. When calculating the fraction of NPP consumed, the uncertainty in NPP estimates dominates over the uncertainty in consumption estimates, and particularly so in logged forest and oil palm.

## Caveats

There are a number of caveats in our analysis. Some taxa are probably under-sampled. These include several small ground-layer insectivorous mammals (in particular, shrews) that cannot be reliably detected using either camera traps or fruit-baited live traps (pitfall traps with drift fence would be required) and 16 frugivorous or nectivorous bat species that are difficult to capture in the tropical forest understorey[55] and are likely to utilize the study area to some degree. Data for fully arboreal mammal species such as primates and flying squirrels were estimated from other studies in the same region (Supplementary Data 1). We did not measure NPP in old-growth forests within the same landscape as the logged forests. However, the variation of measured NPP across old-growth sites in northeast Borneo with very strongly varying soil substrate is fairly small (range 12.03–15.53 Mg C ha$^{-1}$ yr$^{-1}$; that is, ±9%)[14], so we fully expect NPP in old-growth forests in the SAFE landscape to be within this range. Our analysis also assumes no dietary shifts within species across the disturbance gradients (for example, invertebrates make up the same fraction of diet of particular species whether in old-growth or logged forest), but a sensitivity test shows possible dietary shifts have negligible effect (Supplementary Discussion). Finally, we assume a correlation between animal presence and consumption of resources that may not hold in all cases. As noted above, the animals observed in the oil palm plantation may be passing between fragments of logged and/or riparian forest, and hence our estimate of consumption within oil palm is likely to be an overestimate. There are, nonetheless, a small number of animals such as bearded pig, macaques and small carnivores that favour the oil palm as a feeding area, with its abundance of palm fruit and rodents[27,56]. Our logged forest landscape is

adjacent to a larger area of more moderately logged forests to the north (Fig. 1). More isolated and fragmented logged forests might be more defaunated than those studied here, even in the absence of hunting, and would therefore show a smaller increase in energetics.

## Reporting summary

Further information on research design is available in the Nature Portfolio Reporting Summary linked to this article.

## Data availability

The per species energetics data and REM parameters (mammals) are available in Supplementary Data 1 and 2.

## Code availability

The code for processing and statistically analysing the data is available as Supplementary Methods. The REM analysis code is available from the corresponding author on reasonable request or from supplementary methods of ref. [38].

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

**Acknowledgements** This study was part of the SAFE Project, financially supported by the UK Natural Environment Research Council's Human-Modified Tropical Forests Program (NE/K016407/1 and NE/K016377/1), the Sime Darby Foundation and Bat Conservation International. This paper is also a product of the Global Ecosystems Monitoring network (gem.tropicalforests.ox.ac.uk) that provided training and methods. Maliau Basin and Danum Valley Management Committees, Royal Society Southeast Asia Rainforest Research Partnership, Sabah Foundation, Benta Wawasan, the State Secretary, Sabah Chief Minister's Departments, Sabah Forestry Department, Sabah Biodiversity Council and the Economic Planning Unit are acknowledged for their support and access to the sites in Sabah. We thank K. Sam and R. Dunn for helpful advice on parts of the analysis. Y.M. was supported by the Jackson Foundation and a European Research Council Advanced Investigator Grant, GEM-TRAIT (321131). M.J.S. was supported by a Leverhulme Trust Research Leadership Award, and R.M.E. was supported by the NOMIS Foundation.

**Author contributions** Y.M. conceived the analysis and led the writing of the paper with input from T.R., O.R.W., N.J.D., Z.G.D., R.M.E. and M.J.S. O.R.W. developed the REM modelling; O.R.W., N.J.D., S.L.M. and M.J.S. collected and analysed the animal data; T.R. collected and analysed the vegetation data and worked on energetic data analysis. R.M.E. conceived and implemented the SAFE Project. H.B., N.M. and R.N. assisted in collection of field data with supervision from Y.M., Z.G.D., R.M.E. and M.J.S. All authors commented on the draft.

**Competing interests** The authors declare no competing interests.

**Additional information**
**Correspondence and requests for materials** should be addressed to Yadvinder Malhi.

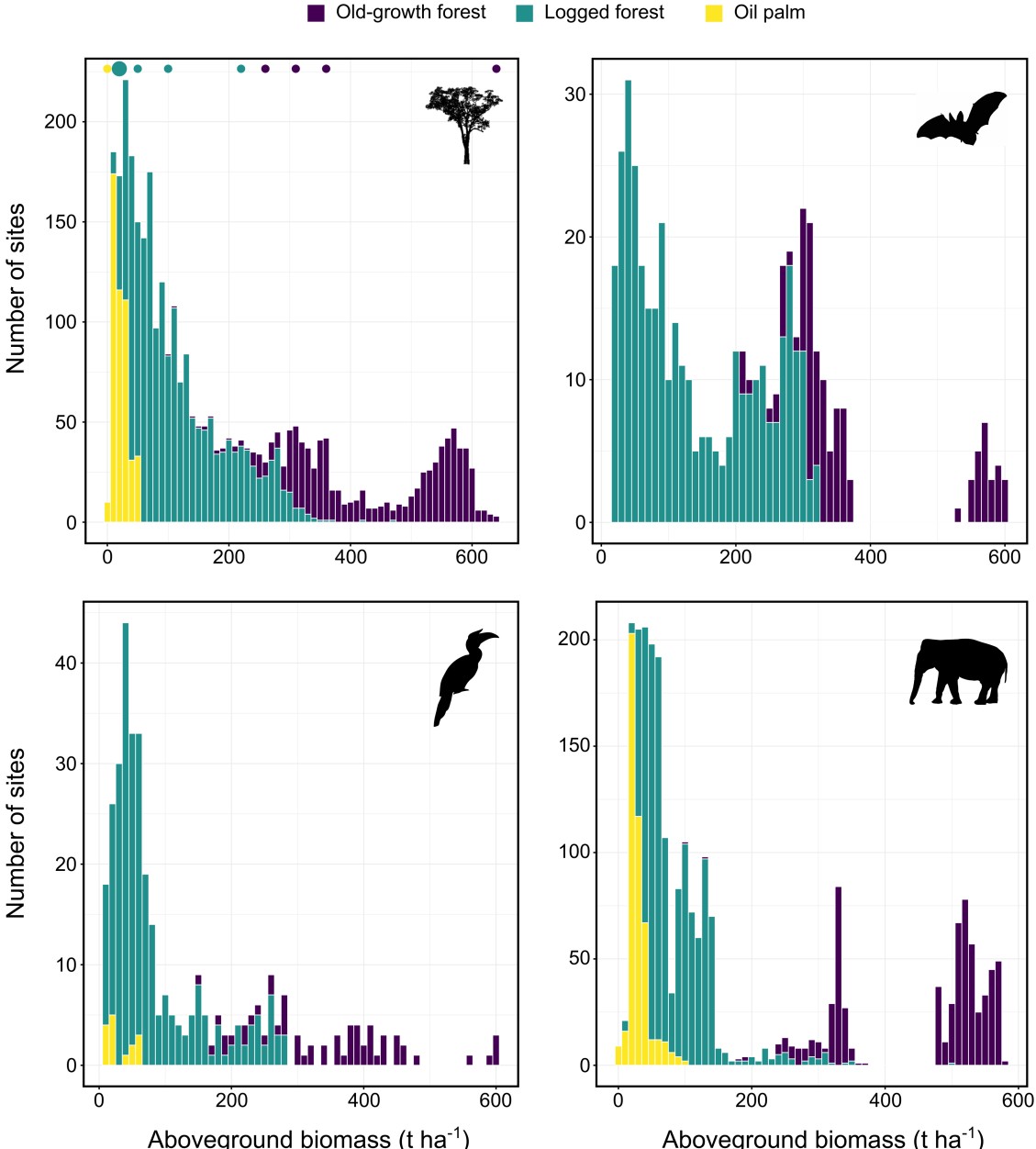

**Extended Data Fig. 1 | Distribution of sampling locations across the gradient of logging intensity in the study landscape, characterised using aboveground dry biomass (t/ha).** We estimate biomass from a spatially-explicit surface of carbon density (30 m resolution) derived from airborne Light Detection and Ranging (LiDAR) data (see[57] for full sampling details) and convert carbon to dry biomass using a conversion factor of 0.47 ([58]). To provide a representative sample of local habitat conditions, biomass was extracted as mean values from 100 m radii buffers around each sampling point. At this resolution there are a broad range of dry biomass values in both old growth and logged forests, but the mean values are clearly distinguished. The histogram top-left shows the biomass and habitat type extracted at each sampling location for all focal taxa combined. The taxa-specific plots simply break that information down to each focal group: the sampling points for vegetation primary productivity (points at top of top-left panel), bats (top-right), birds (bottom-left) and terrestrial mammals (bottom-right) span this gradient well.

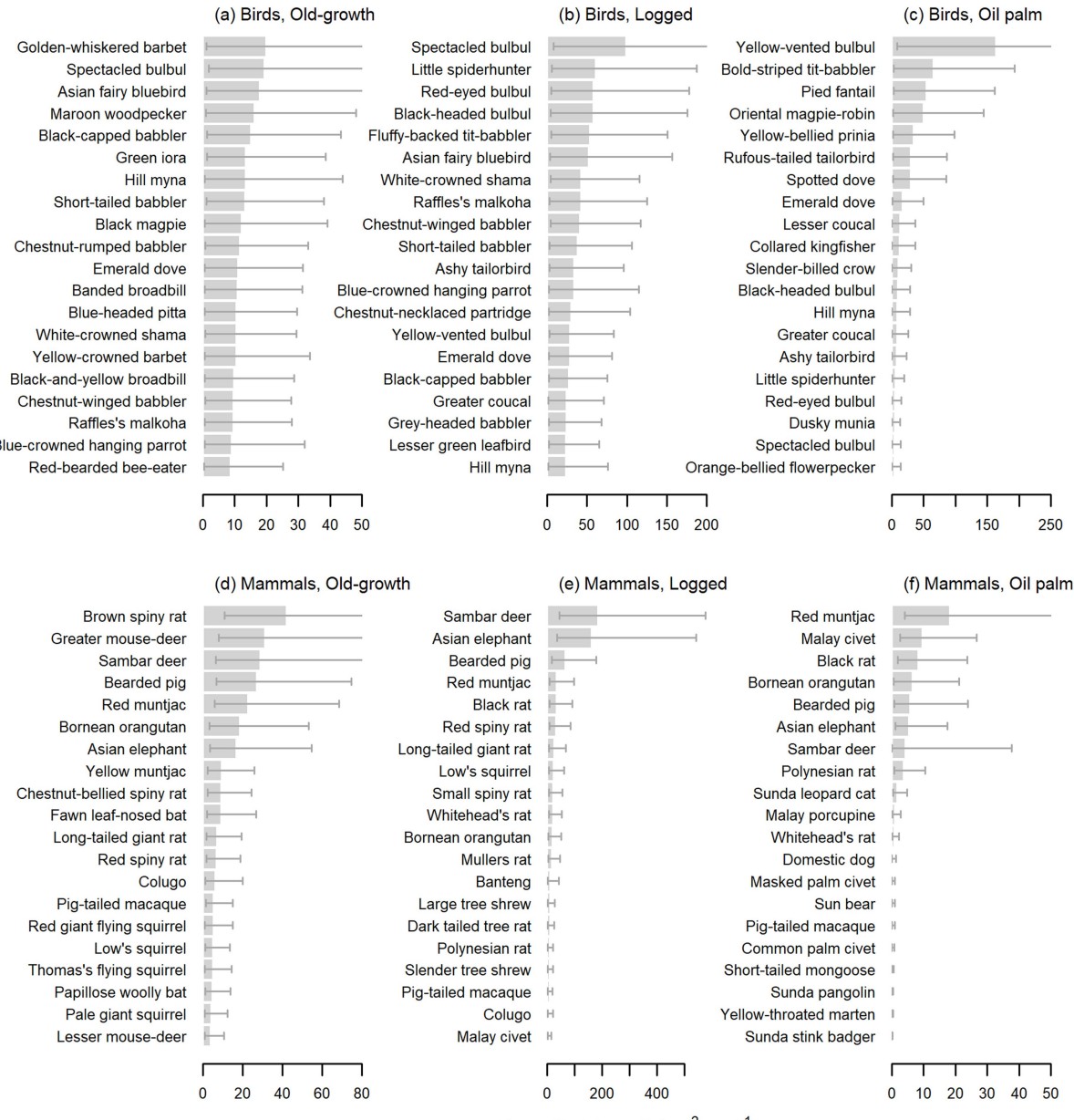

**Extended Data Fig. 2 | The dominant bird and mammal species in terms of food energy consumption in old-growth forest, logged forest and oil palm plantation.** Species-level resource consumption in birds (a-c) and mammals (d-f) (top 20 consumers in each forest type). Error bars denote 95% confidence intervals, derived from 10,000 Monte Carlo simulation estimates incorporating uncertainty in body mass, population density, the daily energy expenditure equation, assimilation efficiency of the different food types and composition of the diet of each species. Data for all species in the study are provided in Supplementary Data 1.

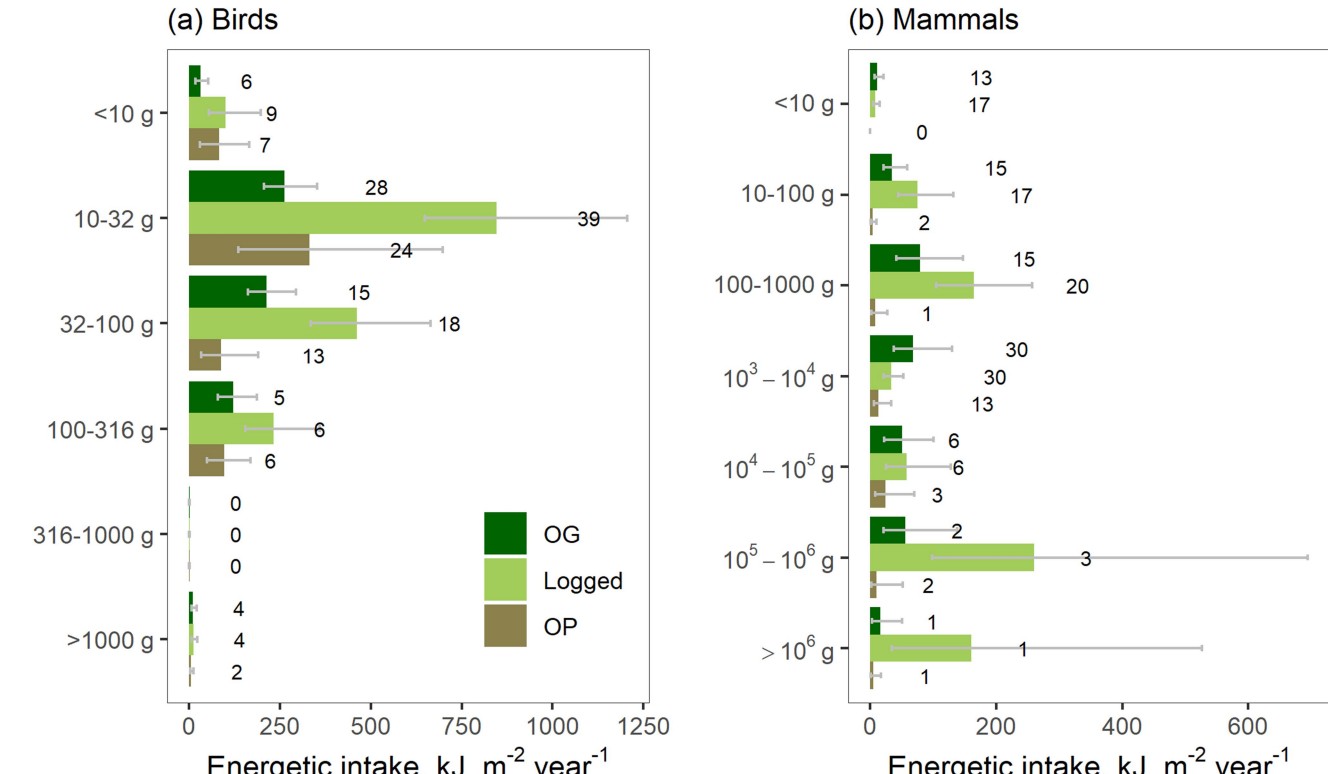

**Extended Data Fig. 3 | Variation of energy intake by birds and mammals by size class.** Direct energetic intake of birds (a) and mammals (b) by body mass class (logarithmic scale) in old growth forest (OG), logged forest and oil palm plantation (OP). The numbers next to the bars indicate the number of species in each class. Error bars denote 95% confidence intervals, derived from 10,000 Monte Carlo simulation estimates incorporating uncertainty in body mass, population density, the daily energy expenditure equation, assimilation efficiency of the different food types and composition of the diet of each species. Data provided in Supplementary Data 4.

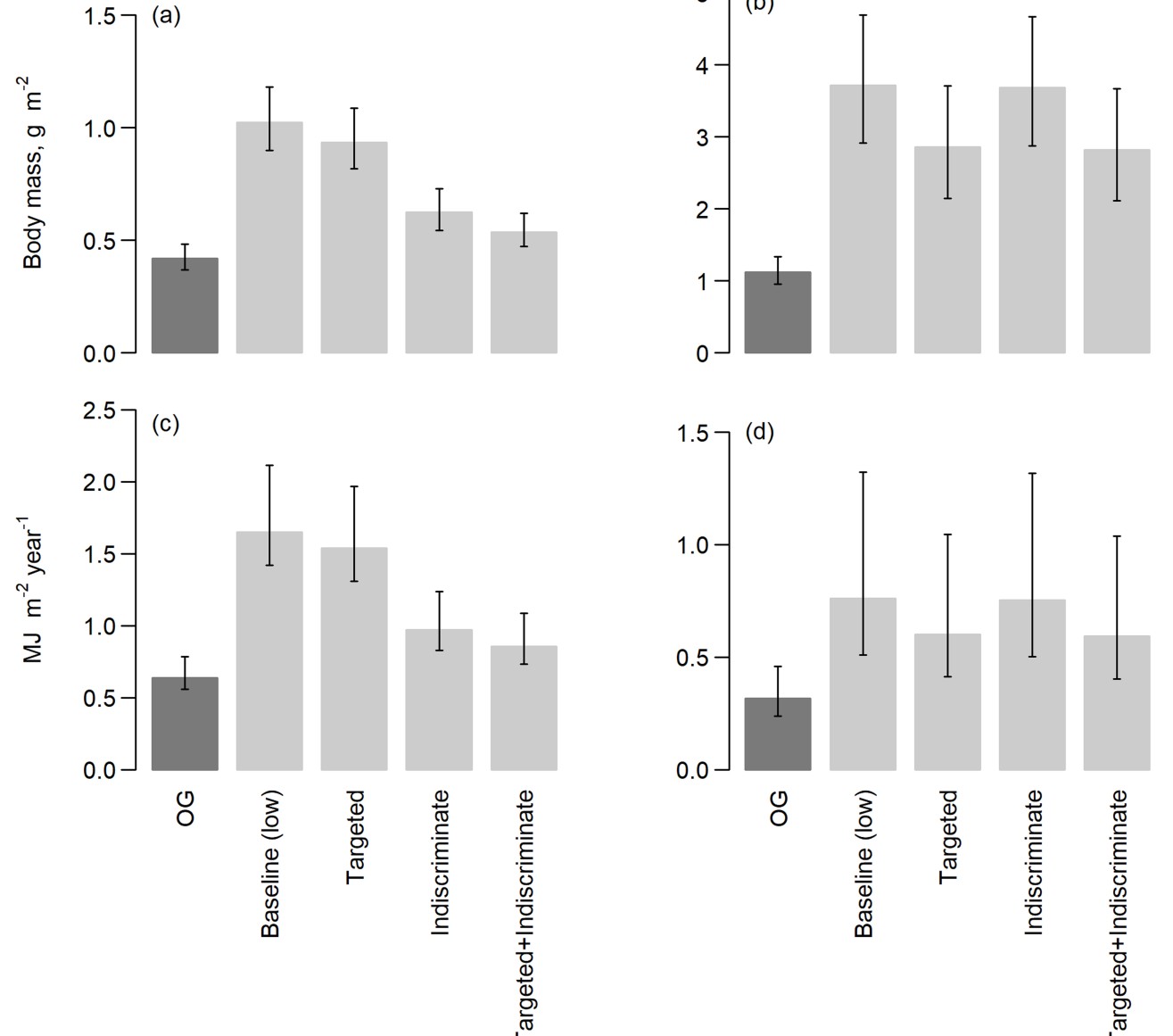

**Extended Data Fig. 4 | Simulations of the effects of hunting on the total biomass and energy intake of birds and mammals.** Body mass of birds (a) and mammals (b) and energetic food intake of birds (c) and mammals (d) in old growth forest (OG, dark grey, no hunting) and in logged forest (light grey) under four different hunting scenarios: observed low hunting pressure (baseline) and simulated 50% reduction in population density of targeted hunted species, indiscriminately hunted species and both targeted and indiscriminately hunted species. Targeted hunted species include commercially valuable birds, and gun-hunted mammals (bearded pig, ungulates, banteng and mammals with medicinal value). Indiscriminately hunted species include birds and mammals likely to be trapped with nets and snares. For the list of species in each category see Supplementary Data 1. Note that this is not an exhaustive analysis of the hunting pressure in the study area but an illustrative estimate of the potential impact of hunting on trophic energetics. Targeted hunted bird species potentially include 13% of bird species, which account for 17% of bird body mass and 14% of bird energy consumption under the observed low hunting pressure. Targeted hunted mammal species potentially include 10% of mammal species, which account for 46% of body mass, 42% of mammal energy consumption under the observed low hunting pressure. Indiscriminately hunted bird species potentially include 72% of bird species, which account for 78% of bird body mass and 82% of bird energy consumption under the observed low hunting pressure. Indiscriminately hunted mammal species potentially include 22% of mammal species, which account for 2% of mammal body mass and 2% of mammal energy consumption. With both hunting pressures applied simultaneously, hunted bird species potentially include 86% of species, 95% of bird body mass and 96% of bird energy consumption under the observed low hunting pressure, and hunted mammal species potentially include 32% of mammal species, 48% of mammal body mass and 44% of mammal energy consumption under the observed low hunting pressure. Data provided in Supplementary Data 6.

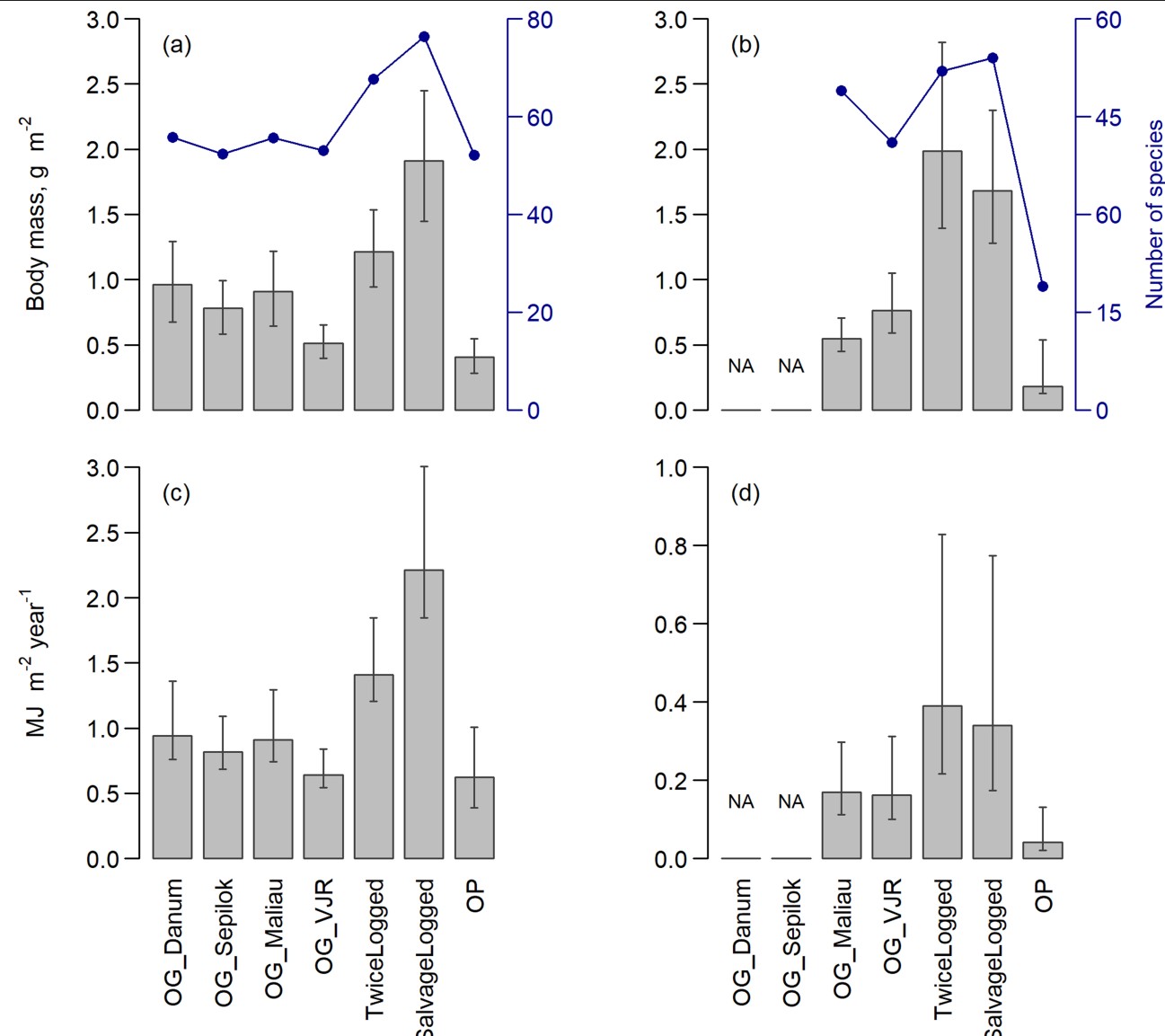

**Extended Data Fig. 5 | Variation of bird and mammal biomass and energy intake across individual old-growth forest sites and logging intensities.** Body mass and species richness of birds (a) and mammals (b) and energetic food intake of birds (c) and mammals (d) across old growth forests (OG), logged forests and oil palm plantations (OP). OG forest data were analysed separately by four OG sites for birds and two sites for mammals (see Fig 1 for map), and the logged forest data were split into twice logged and heavily logged areas. For mammals, only species studied using camera traps and harp traps were included (63%, 63% and 77% of mammal species, and 53%, 45% and 63% of total energetic food intake in OG, logged forest and OP, respectively). Error bars are 95% confidence intervals derived from 10,000 Monte Carlo simulation estimates incorporating uncertainty in body mass, population density, the daily energy expenditure equation, assimilation efficiency of the different food types and composition of the diet of each species. Data provided in Supplementary Data 7.

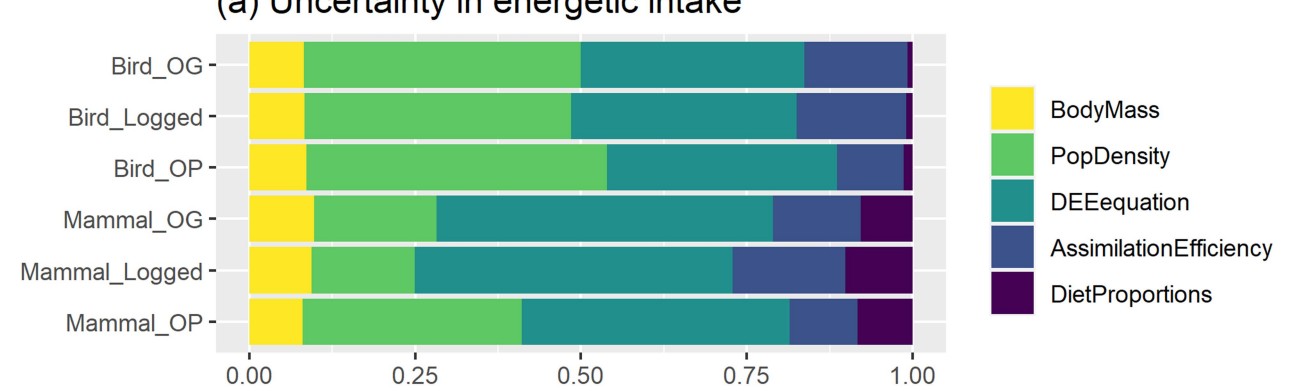

## (a) Uncertainty in energetic intake

## (b) Uncertainty in % NPP consumed

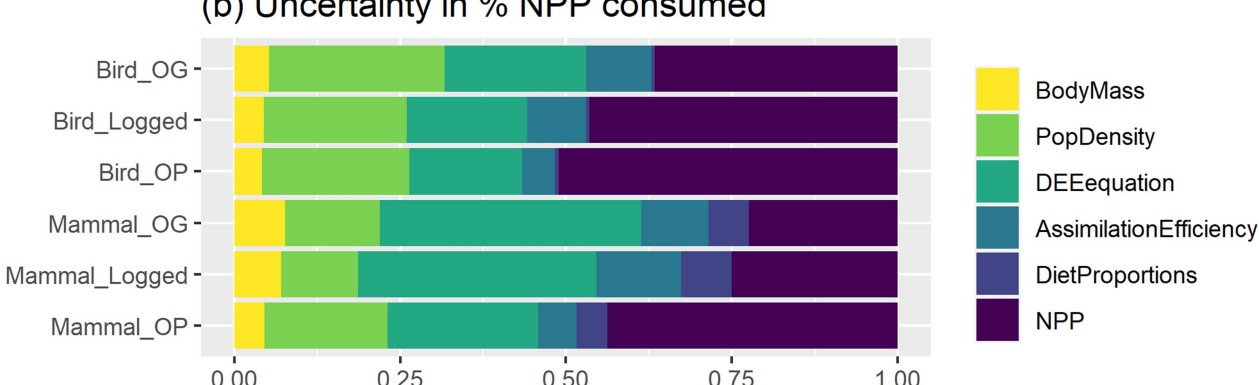

**Extended Data Fig. 6 | Sources of uncertainty in estimation of energetic intake.** Sources of contribution to uncertainty in energetic intake (a) and proportion of net primary productivity (NPP) consumed (b) for birds and mammals across the habitat types of old growth forest (OG), logged forest and oil palm plantation (OP). We assumed there was uncertainty in the following variables: body mass of species, population density, the daily energy expenditure (DEE) equation, assimilation efficiency of the different food types, fractional composition of the diet of each species, and NPP. Uncertainty estimates were derived from 10,000 Monte Carlo simulations, and the contribution of each variable to the total uncertainty was assessed by running the simulations assuming uncertainty in all variables simultaneously and in one variable at a time. Data provided in Supplementary Data 5.

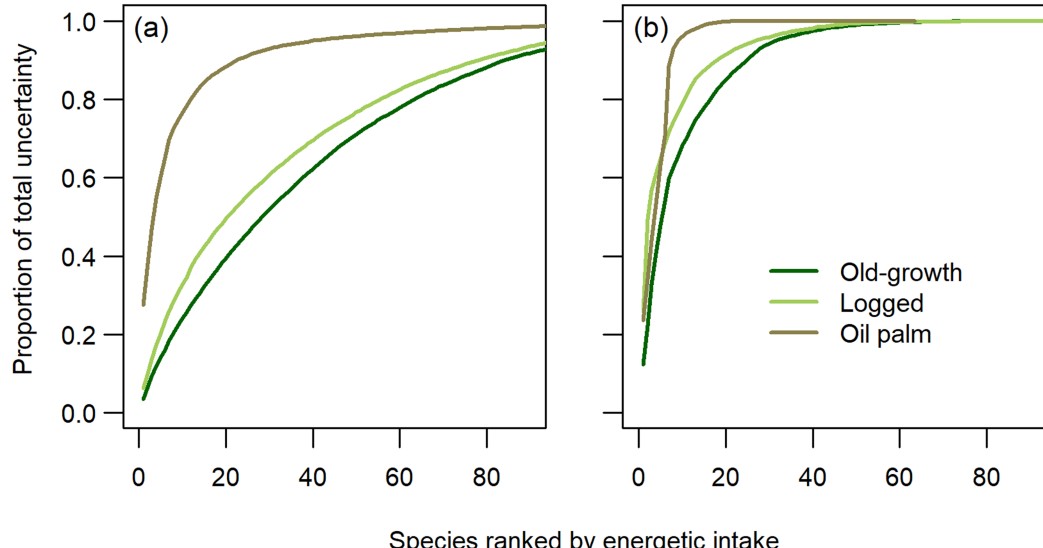

**Extended Data Fig. 7 | The proportion of uncertainty in total energetic uptake contributed by individual bird or mammal species.** The proportion of total uncertainty contributed by each species ranked by energy consumption for birds (a) and mammals (b).

**Extended Data Table 1 | Energetic food intake (kJ m$^{-2}$ year$^{-1}$) of different food types by birds and mammals and 95% confidence intervals of the estimates across the disturbance gradient from old growth forest through logged forest to oil palm**

| | | Fruit | Seeds/ nuts | Nectar | Leaves | Invertebrates | Vertebrates | Total |
|---|---|---|---|---|---|---|---|---|
| Old growth | Birds | 165.4 | 19 | 21 | 0 | 416.9 | 13.9 | 636.2 [556.4, 782.8] |
| | Mammals | 96.4 | 60.3 | 0.1 | 75.7 | 82.4* | 1.3 | 316 [238.5, 458.5] |
| | Total | 261.8 | 79.3 | 21.1 | 75.5 | 499.3 | 15.2 | 952.2 [844.4, 1162.5] |
| Logged | Birds | 445.6 | 47.3 | 103.4 | 0 | 1036.9 | 17.4 | 1650.6 [1421.1, 2114.4] |
| | Mammals | 227.3 | 118.3 | 0.3 | 288.2 | 123.6* | 3.9 | 761.6 [509.5, 1321.6] |
| | Total | 672.9 | 165.6 | 103.7 | 288.2 | 1160.5 | 21.2 | 2412.1 [2076.8, 3133.8] |
| Oil Palm | Birds | 116.8 | 57 | 11.4 | 0 | 399.4 | 17.4 | 602.1 [370.9, 948.0] |
| | Mammals | 19.8 | 11.1 | 0 | 17.2 | 12 | 6.1 | 66.2 [39.9, 131.8] |
| | Total | 136.6 | 68.1 | 11.4 | 17.2 | 411.4 | 23.5 | 668.3 [438.6, 1063.7] |

Numbers with an asterisk include estimates for some bats, values without the bat estimates are 47.7 kJ m$^{-2}$ year$^{-1}$ in old growth and 99.3 kJ m$^{-2}$ year$^{-1}$ in logged forest. The confidence intervals are derived from 10,000 Monte Carlo simulation estimates incorporating uncertainty in body mass, population density, the daily energy expenditure equation, assimilation efficiency of the different food types, and composition of the diet of each species.

**Extended Data Table 2 | Proportion of NPP (%), and 95% confidence intervals of the estimates, consumed by food type across the disturbance gradient from old growth forest (OG) through logged forest to oil palm**

|  |  | Fruit | Seeds/ nuts | Nectar | Leaves | Invertebrates | Vertebrates | Total |
|---|---|---|---|---|---|---|---|---|
| **Old growth** | **Birds** | 0.282 | 0.032 | 0.036 | 0 | 0.711 | 0.024 | 1.085<br>[0.902, 1.435] |
|  | **Mammals** | 0.164 | 0.103 | 0.0002 | 0.129 | 0.141* | 0.002 | 0.539<br>[0.389, 0.816] |
|  | **Total** | 0.446 | 0.23 | 0.036 | 0.129 | 0.852 | 0.03 | 1.624<br>[1.354, 2.132] |
| **Logged** | **Birds** | 0.62 | 0.066 | 0.144 | 0 | 1.443 | 0.024 | 2.297<br>[1.764, 3.451] |
|  | **Mammals** | 0.316 | 0.165 | 0.0004 | 0.401 | 0.172* | 0.005 | 1.060<br>[0.663, 1.995] |
|  | **Total** | 0.936 | 0.23 | 0.144 | 0.401 | 1.615 | 0.03 | 3.357<br>[2.573, 5.071] |
| **Oil Palm** | **Birds** | 0.155 | 0.076 | 0.015 | 0 | 0.53 | 0.023 | 0.799<br>[0.404, 2.132] |
|  | **Mammals** | 0.026 | 0.015 | 0 | 0.023 | 0.016 | 0.008 | 0.088<br>[0.045, 0.267] |
|  | **Total** | 0.181 | 0.09 | 0.015 | 0.023 | 0.546 | 0.032 | 0.886<br>[0.470, 2.236] |

Numbers with an asterisk include estimates for some bats, values without the bat estimates are 0.081% in old growth and 0.138% in logged forest. The confidence intervals are derived from 10,000 Monte Carlo simulation estimates incorporating uncertainty in body mass, population density, the daily energy expenditure equation, assimilation efficiency of the different food types, composition of the diet of each species, and NPP.

# Reporting Summary

## Statistics

For all statistical analyses, confirm that the following items are present in the figure legend, table legend, main text, or Methods section.

| n/a | Confirmed | |
|---|---|---|
| ☐ | ☒ | The exact sample size (*n*) for each experimental group/condition, given as a discrete number and unit of measurement |
| ☐ | ☒ | A statement on whether measurements were taken from distinct samples or whether the same sample was measured repeatedly |
| ☐ | ☒ | The statistical test(s) used AND whether they are one- or two-sided<br>*Only common tests should be described solely by name; describe more complex techniques in the Methods section.* |
| ☐ | ☒ | A description of all covariates tested |
| ☐ | ☒ | A description of any assumptions or corrections, such as tests of normality and adjustment for multiple comparisons |
| ☐ | ☒ | A full description of the statistical parameters including central tendency (e.g. means) or other basic estimates (e.g. regression coefficient) AND variation (e.g. standard deviation) or associated estimates of uncertainty (e.g. confidence intervals) |
| ☐ | ☒ | For null hypothesis testing, the test statistic (e.g. $F$, $t$, $r$) with confidence intervals, effect sizes, degrees of freedom and $P$ value noted<br>*Give P values as exact values whenever suitable.* |
| ☒ | ☐ | For Bayesian analysis, information on the choice of priors and Markov chain Monte Carlo settings |
| ☐ | ☒ | For hierarchical and complex designs, identification of the appropriate level for tests and full reporting of outcomes |
| ☐ | ☒ | Estimates of effect sizes (e.g. Cohen's *d*, Pearson's *r*), indicating how they were calculated |

*Our web collection on statistics for biologists contains articles on many of the points above.*

## Software and code

Policy information about availability of computer code

| Data collection | No software was used in data collection |
|---|---|
| Data analysis | Data was analysed and visualised with a custom code using R (4.2.0 and earlier) and the following R packages: ggplot2, ggbreak, ggpubr, cowplot, viridis, nlme, reshape2, TruncatedNormal, and vegan. |

For manuscripts utilizing custom algorithms or software that are central to the research but not yet described in published literature, software must be made available to editors and reviewers. We strongly encourage code deposition in a community repository (e.g. GitHub). See the Nature Portfolio guidelines for submitting code & software for further information.

## Data

Policy information about availability of data

All manuscripts must include a data availability statement. This statement should provide the following information, where applicable:

- Accession codes, unique identifiers, or web links for publicly available datasets
- A description of any restrictions on data availability
- For clinical datasets or third party data, please ensure that the statement adheres to our policy

The per-species energetics data, and REM model parameters (mammals) are available in Supplementary Data Tables 1 and 2.

March 2021

## Human research participants

Policy information about studies involving human research participants and Sex and Gender in Research.

| | |
|---|---|
| Reporting on sex and gender | NA |
| Population characteristics | NA |
| Recruitment | NA |
| Ethics oversight | NA |

Note that full information on the approval of the study protocol must also be provided in the manuscript.

# Field-specific reporting

Please select the one below that is the best fit for your research. If you are not sure, read the appropriate sections before making your selection.

☐ Life sciences   ☐ Behavioural & social sciences   ☒ Ecological, evolutionary & environmental sciences

For a reference copy of the document with all sections, see nature.com/documents/nr-reporting-summary-flat.pdf

# Ecological, evolutionary & environmental sciences study design

All studies must disclose on these points even when the disclosure is negative.

| | |
|---|---|
| Study description | *Briefly describe the study. For quantitative data include treatment factors and interactions, design structure (e.g. factorial, nested, hierarchical), nature and number of experimental units and replicates.* |
| Research sample | *Describe the research sample (e.g. a group of tagged Passer domesticus, all Stenocereus thurberi within Organ Pipe Cactus National Monument), and provide a rationale for the sample choice. When relevant, describe the organism taxa, source, sex, age range and any manipulations. State what population the sample is meant to represent when applicable. For studies involving existing datasets, describe the data and its source.* |
| Sampling strategy | *Note the sampling procedure. Describe the statistical methods that were used to predetermine sample size OR if no sample-size calculation was performed, describe how sample sizes were chosen and provide a rationale for why these sample sizes are sufficient.* |
| Data collection | *Describe the data collection procedure, including who recorded the data and how.* |
| Timing and spatial scale | *Indicate the start and stop dates of data collection, noting the frequency and periodicity of sampling and providing a rationale for these choices. If there is a gap between collection periods, state the dates for each sample cohort. Specify the spatial scale from which the data are taken* |
| Data exclusions | *If no data were excluded from the analyses, state so OR if data were excluded, describe the exclusions and the rationale behind them, indicating whether exclusion criteria were pre-established.* |
| Reproducibility | *Describe the measures taken to verify the reproducibility of experimental findings. For each experiment, note whether any attempts to repeat the experiment failed OR state that all attempts to repeat the experiment were successful.* |
| Randomization | *Describe how samples/organisms/participants were allocated into groups. If allocation was not random, describe how covariates were controlled. If this is not relevant to your study, explain why.* |
| Blinding | *Describe the extent of blinding used during data acquisition and analysis. If blinding was not possible, describe why OR explain why blinding was not relevant to your study.* |

Did the study involve field work?   ☒ Yes   ☐ No

## Field work, collection and transport

| | |
|---|---|
| Field conditions | Tropical rainforest (intact and logged) and oil palm plantations |
| Location | Data from logged forests were collected across the Stability of Altered Forest Ecosystems (SAFE) Project Landscape (4° 43' N, 117° 35' E) in Sabah, Malaysia 30, a lowland mosaic landscape of logged forest and oil palm plantation. Data for oil palm plantations were |

collected from adjoining oil palm estates. Data for the old growth forests were collected from the Braintian-Tantulit Virgin Jungle Reserve (VJR, a large fragment adjoining the logged forest landscape), and also in three other old growth forest reserves in Sabah, the Maliau Basin Conservation Area (vegetation, birds and mammals), the Danum Valley Conservation Area (vegetation and birds) and Sepilok Forest Reserve (birds only).

| Access & import/export | Permits was given to work in all field sites. No sample export was required. |
| Disturbance | The work caused minimal disturbance and exploited logging and palm oil plantations that were already established. |

# Reporting for specific materials, systems and methods

We require information from authors about some types of materials, experimental systems and methods used in many studies. Here, indicate whether each material, system or method listed is relevant to your study. If you are not sure if a list item applies to your research, read the appropriate section before selecting a response.

## Materials & experimental systems

| n/a | Involved in the study |
|-----|------------------------|
| ☒ | Antibodies |
| ☒ | Eukaryotic cell lines |
| ☒ | Palaeontology and archaeology |
| ☒ | Animals and other organisms |
| ☒ | Clinical data |
| ☒ | Dual use research of concern |

## Methods

| n/a | Involved in the study |
|-----|------------------------|
| ☒ | ChIP-seq |
| ☒ | Flow cytometry |
| ☒ | MRI-based neuroimaging |

