## [Peer Review File · Nature]

Manuscript Title: Logged tropical forests have amplified and diverse ecosystem energetics

Reviewer Comments & Author Rebuttals

Reviewer Reports on the Initial Version:

Referees' comments:

Referee #1 (Remarks to the Author):

Summary of the key results:

The manuscript "Logged tropical forests have amplified and diverse ecosystem energetics" by Malhi et al investigates which part of the primary productivity is consumed by fauna and how this consumption varies among old-growth forests, logged forests and oil palm plantations in Borneo. By using datasets gathered through an amazing sampling effort (~42k camera trap-nights), they found that logged forests have amplified energy flow through consumption relative to old-growth and palm tree forests.

Originality and significance:

Their work addresses novel questions. Although previous work has compared mammalian and bird communities across gradients of land-use change including old-growth forests, logged forests, and/or oil palm plantations (e.g. Wearn et al. 2017 - Biol. Conserv.; Costantini et al. 2016 - Biol. Conserv.; Wearn et al. 2016 – Ecol. App., Wearn et al. 2018 - J. App. Ecol), I am not aware of any previous research adopting an energetic approach to compare different forest classes in tropical regions. I consider this novel aspect, quality of the datasets and quality of the presentation as key strengths of their research.

Data, methodology & appropriate use of statistics and treatment of uncertainties:

There are, however, some methodological points that may need some clarification. For example:

1 - Would differences in sampling effort across their land-use classes affect their results?

Sampling effort in logged forests was 1.7 higher than in old-growth forests and 2.6 higher than in oil palm plantations.

2 - Were their logged study sites distributed across a gradient of logging intensities and what type of logging operations were conducted there?

Reduced-impact logging and conventional logging result in distinct impacts for tropical biodiversity (e.g. Bicknell et al. 2014 – Current Biology). Moreover, logging intensities vary across areas within a concession and oversampling low-impacted regions could bring extra confounding effects that may affect their biodiversity assessments and energy flow estimations.

3 - How is the landscape context of their work and could time since logging be driving their outcomes?

A higher amount of old-growth forest [source populations] in the landscape may explain the lack of differences in bird diversity. Alternatively, if old-growth forests have been impacted by other

disturbances (e.g. hunting, fires, illegal logging), it could be that shifted baselines would influence energetic values observed in this forest class. On the other hand, the 40-46 years interval between the first round of logging (the 1970s) and their surveys (between 2010-2016) could mean that results reflect the reality of logged forests after many years of recovery -- and this [time since logging operations, post-logging interventions and inactive concessions – e.g. Burivalova et al. 2020 – Biol. Conserv; Cerullo et al. 2019 – For. Ecol. Manag.] is not recognised in their caveats.

Conclusions:

Despite my concerns highlighted above (and below in the minor comments), the study and approaches seem robust and valid. Further suggestions and clarifications on my concerns are provided below.

Minor comments:

L14: Declines in logged forests have been observed even for non-specialist species (depending on logging intensities and approach adopted – e.g. Burivalova et al. 2014 – Current Biology, França et al. 2017 – Biol. Conservation). Would be interesting if the authors could recognise that somewhere in the text.

L27-28: Is there any available data and/or references that could support this claim of the importance of invertebrates as a substantial constrain on vegetation growth rates?

L37-38: Low ecological value compared to which baseline?

I would restructure the second part of this sentence. Although studies have shown that human-modified forests have a higher ecological value when compared with deforested areas for agriculture (without claiming that human-modified forests/ecosystems are suitable for conversion to other land uses) there is a large body of research showing the importance of pristine /undisturbed ecosystems for maintaining biodiversity (high ecological value) when compared with human-modified ecosystems (low ecological value) – e.g. Gibson et al. 2011 – Nature; Barlow et al. 2016 - Nature).

L42: Not only old-growth specialist species... overall biodiversity loss have been documented in logged forests at the community level and associated ecological functions for distinct taxa and multiple (forest specialist and non-specialist species) – e.g. Burivalova et al. 2014 – Current Biology; Burivalova et al. 2015 – Proc. R. Soc. B, França et al. 2017 – Biol. Conservation.

L58-61: Any references on how the energetic pathways can be linked to ecosystem processes and trophic factors?

L78: An amazing sampling effort!

L82: What “34 site-years” mean? I recommend some clarification on this.

L88-89: Would the maintenance of bird diversity be related to (a) shifted baselines of old-growth forests and/or (b) landscape context (e.g. amount of old-growth forest in the region)? Have you accounted for these (a,b) during your models/uncertainty calculations?

L142-147: Any references available for these claims?

L194 and L263-266: Are there any information on the (a) logging intensity and (b) time since logging at the study site level?

Time since logging is a key factor driving ecological patterns in logged forests. When were the subsequent rotations?

Logging intensity is not uniform across concessions, but is also a key driver of ecological patterns in tropical forests (e.g. Burivalova et al. 2014 – Current Biology, França et al. 2017 – Biol. Conservation).

Do the authors have access to logging intensity values for study sites?

Assessing data on these (a & b) is key when contrasting these two forest classes and lack of differences could be a result of low logging intensity and/or shifted baselines. There is information provided in L263-266, but these seem more like averaged values across the sites. Yet, the ground reality of logging concessions sometimes means that some sites /regions in the concession have a higher volume of timber extraction compared to others sites/regions. Thus, knowing whether surveyed sites have been distributed across a gradient of logging intensities would be helpful to understand if there were any biases towards low- or highly-impacted forests.

L267-268: I would expect logging roads to facilitate access to hunters with further impacts particularly for large-bodied terrestrial birds.

L292-293: Sampling effort in logged forests was 1.7 higher than in old-growth forests and 2.6 higher than in oil palm plantations. Would that be a factor driving /affecting the study outcomes?

I mean, higher sampling efforts may result in more detections /density estimations.

L306-307: Would that affect their comparisons/results?

L322-323: I am not a specialist in camera trap work and was wondering if the REM approach accounts for the same individual passing through the (same or different) cameras in a given territory / region and how this could influence density estimations. Would be possible to provide some clarification on this?

L341-342: Any references for this?

L363-364: If each site received 4 visits, would that mean that the maximum detection would be four? Or would multiple individuals be accounted in a single sampling occasion? Clarifying this would be very helpful.

L400-410: It would be interesting if uncertainty calculation accounted for differences in sampling efforts across the land-use classes that were compared.

L442: There is an extra parenthesis here.

Referee #2 (Remarks to the Author):

This is a highly innovative and very detailed analysis of shifts in ecosystem energetics in tropical forests after logging. It is unique and innovative because it is the first analysis of shifts in energy flows due to logging, and provides a first and new assessment of the functioning of logged tropical forests. The study provides a very detailed account of consumption pathways across trophic levels, making use of sophisticated empirical data on changes in primary productivity and abundance of animal groups between logged and unlogged forests. By comparing plots in a logged and an unlogged forest area in Borneo, the authors show that NPP and animal abundance is higher in logged forests, and so is the energy intake by bird and mammals.

The study changes the way we perceive logged tropical forests and their ecology: from ecosystems that are functionally and structurally “degraded” and have lost species and carbon; to ecosystems that – in spite of structural alteration – possess a much higher productivity and faunal abundance, exhibit enhanced energetic pathways, and resilient energy flows. Given that a large share and a large area (400 mln ha) of tropical forests are used for timber production, the paper provides a highly relevant message on an “alternative” way to see these forests: as fully functional ecosystems with a potentially high level of resilience.

I have three concerns about the study:

1. Dietary shifts. The calculation of changes in energy flows between forest types is solely based on differences in abundances (which – to my knowledge – are estimated appropriately), assuming that logging does not cause shifts in diet or feeding activity of bird or mammal species. I wonder to what extent results are robust to possible changes in feeding behaviour and trophic level (diet). Although this is not my field of expertise, I found an example of a study documenting strong dietary shifts (as much as one trophic level) of birds in logged vs undisturbed forests (Hamer et al 2015). The shift in logged forests to a higher abundance of pioneer tree species, understorey shrubs/herbs and thus palatable leaves may increase insect herbivory (which unfortunately was not included in the measurements) and hence shift diet of birds and mammal species. While I acknowledge and appreciate that shifts in diet were to some extent treated in the uncertainty analyses, I do not think that this analysis allows to evaluate the effects of a pervasive shift in diet. I therefore suggest the authors conduct a robustness test to evaluate whether conclusions are robust to observed (see Hamer et al 2018) or expected dietary shifts due to logging.

2. Oil palm. I do not see the need or added value of including the results of the oil palm plantation in this manuscript. The text is now framed in relation to selective logging in tropical forests, the main messages are about logging effects on energy flows, and the main innovation is in the quantification of logging effects on energy flows. In my view, there is no reason to include the oil palm results, also not as a contrast with other forest types. Including oil palm may only divert attention from the main messages on logging effects. I therefore recommend leaving out the oil palm results.

3. Representativeness. This study was done in one region in Malaysia, with high logging intensity (compared to Neotropical and African logging intensities) and low hunting pressure (compared to most logged forests, probably). I missed a discussion on the representativeness of this study for other logged tropical forests, both within and outside SE Asia. To what extent are similar shifts in NPP, faunal abundance and energy flows expected at lower logging intensity? The reported NPP shifts can be compared to similar studies in other regions (GEM studies in Brazil; other studies on woody productivity in logged forests), to provide context. To what extent are the reported increase

in faunal abundance expected at higher (and more representative) hunting intensity? The study site can be easily put in context using defaunation indices. Finally, the particularly low hunting pressure in the study region also requires some discussion of the expected shifts in ecosystem energetics under higher hunting intensity.

Other comments:

- Abstract. At present, the abstract (second sentence) does not convey the main gap in knowledge that is addressed in the study. Also, I thought that the expression that a “lens shines a new light” was a bit strange: lenses allow light to pass, but do not actively shine... I certainly do like the term “energetics lens”, as this is what the study does: helping us to view logged forests differently. Third, I think it is important to stress that this study was conducted in an area with low hunting pressure, implying that the reported enhanced energetics pathways in logged forests may be jeopardized by high levels of hunting.
- Figure 1. I found the layout of this Figure to be somewhat unorganized and think the quality can be improved (to the standard of Figure 2). It would be logical to start (upper panels) with the NPP and probably with animal densities, and then move down to energy flows. In the present Figure the NPP is somewhat hidden in the middle. Also, I do not get why energetic intake is expressed in kg biomass, while all other productivity measures are expressed as kJ. (One of the claims the authors make is that ecosystem energetic studies allow comparisons using a common metric (energy, kJ), but here this does not seem to be the case.) Finally, it is unclear what the whiskers mean.
- The results of the uncertainty analyses did not make it to the main text, and I would argue that this is a miss. These are important analyses, skilfully conducted and too important to be dealt with in the Methods section alone.
- I could not find results of the mixed effects models that are mentioned in the Methods section.

Reference:

Hamer, K. C., et al. (2015). "Impacts of selective logging on insectivorous birds in Borneo: The importance of trophic position, body size and foraging height." *Biological Conservation* 188: 82-88.

Referee #3 (Remarks to the Author):

This manuscript addresses differences in energetic pathways through plant-bird-mammal communities across undisturbed and logged tropical forests and palm plantations. Data on NPP and densities of birds and mammals on a Borneo field site are used to estimate consumption rates. The study is coming to the conclusion that “Far from being degraded ecosystems, even heavily logged forests can therefore be vibrant and diverse ecosystems with enhanced levels of ecological function.” Overall, I found the topic of the study intriguing and important, and the manuscript is very well written. However, I have concerns that the methods applied can be used to support the conclusions. Despite this critical feedback on the methods and the presentation of the results, I hope that my points can help to strengthen the manuscript.

1) While the conclusions are drawn for degraded ecosystems in general, I have the impression from the methods that just one logged forest has been studied. I might be wrong here but the methods are not very explicit about the sampling site(s?), and such information would be crucially important to judge the level of generality that can be achieved. I certainly understand the constraints of field sampling but if my understanding is correct that this is just one area of logged forest compared to another area of pristine forest, I would be more careful with the conclusions as the results could also just represent one specific case.

2) I am missing a section describing the statistical approaches. How did the authors compare the energy pathways across the three forest types?

3) What is represented by the error bars in the figures (e.g. Fig. 1)?

4) Have the authors considered spatial autocorrelation in their statistical analyses? The description of the study sites is a bit vague but I am under the impression that the logged forest is at one site and the unlogged forests at another site. If this holds true the results are challenged by the alternative hypothesis that spatial effects are driving the differences in densities and thereby energetics. The statistical analyses should test this hypothesis to show that it can be discarded.

5) From the methods description, it appears that the NPP measurements, the mammal and bird surveys have been carried out at different locations. Again, the description is not detailed enough here to allow a thorough evaluation but NPP has been assessed at 5 (logged forest) and 4 (old-growth forest) sites. Camera traps have been set up at 882 locations. Terrestrial mammals have been caught at 1488 locations. Bats have been captured at 42 sampling locations. Avian point counts have been carried out at 356 locations. This description suggests that the data have been obtained at different locations, but there is no description how the data have been grouped or combined to allow for energetic analyses. In addition, the samples have been obtained in different years making it even harder to compare them. The description of the sampling design needs to be given with much more detail on the sampling sites and how the data have been pooled. Overall, however, I am afraid that I am not convinced that such data can be used to estimate energy flows through pathways.

6) I am skeptical that the estimates of energy demands via allometric equations can be used to predict energy flows. They are certainly an important component as they constrain the metabolic rates of the animals but the real-world demand is also affected by environmental factors such as the ambient temperature and interactions with other species. At least, a model predicting energy flows should include effects of temperature (along with allometries) and also the consumption loss rates of the populations. The latter point is based on the pattern that populations can be driven to low densities by predation despite high production rates (and thereby high rates of consumption and energy flow). A model such as the one used in this study ignores this possibility and is thereby too simplistic.

7) In addition, I am also not convinced that a study based on plants, mammals and birds can be used to estimate the energy flows through a community. The vast majority of energy flows goes through invertebrates that dominate most communities in general and tropical communities in particular in

not only their biomass but also their per unit biomass consumption rates. This point is troubling me as the study results are used for very broad conclusions such as "The total flow of energy flux through consumption is amplified across all energetic pathways by 147% in logged forest relative to old-growth forest." This conclusion is drawn despite the statement "Overall, up to 50% of ecosystem net primary production may be supporting the bird and mammal communities in logged forests, predominantly via invertebrate pathways, and herbivory by invertebrates may be a substantial constraint on vegetation growth rates." I am afraid that I am really not convinced that broad statements on energy fluxes can be supported by a study on plants and vertebrates.

Referee #4 (Remarks to the Author):

Summary of the key results

This paper quantifies energetic pathways from trees to birds and mammals in three of the major ecosystem types in Sabah, Malaysia: old-growth forests, selectively logged forests and oil palm plantations. The results show that bird diversity (in terms of species richness and consumption pathways) remains high across the landscape, while mammal diversity is more sensitive to land-use conversion, and declines sharply in oil palm plantations. Total bird and mammal biomass and energy consumption are highest in logged forests, contrary to the common perception that old-growth forests have a higher abundance of fauna than logged forests disturbed by humans.

Originality and significance

In my opinion, the most significant result of this study is the high ecological functionality of selectively logged forests (under low hunting pressure), that support a high abundance and diversity of birds and mammals. Some of these conclusions were already supported by previous results which include some of the data presented in this study: for example, Wearn et al. (2017; cited in this manuscript) showed that mammal abundance was higher in logged forests than in old-growth forests and oil palm plantations. The originality of this new study is the use of energetic pathways that allows an integrative ecosystem-wide approach including most of the food chain of these ecosystems, from primary plant productivity to top predators. It thus provides a good overview of how land use conversion affects ecological interactions in this tropical landscape. The results would therefore be of significance to a broad scientific community, from landscape ecologists to conservationists.

Data & methodology: validity of approach, quality of data, quality of presentation

I have little knowledge in fauna surveys (camera traps, etc.) and the models to infer population densities from such data, so it would be good to have the opinion of someone with more expertise on this. However, all the methods used have all been published in previous articles cited in

references, and I didn't detect any major methodological flaws.

The figures presenting the data are easy to understand and well designed. I have some reservations about Figure 3, which presents data that is less important for the main message and is only quoted once in the main text. I therefore suggest moving Figure 3 to the supplementary file, but of course I leave this decision to the authors.

Some suggestions to make the data provided more in line with the 'FAIR' principles:

- I would suggest putting the supplementary data 1 in an online repository (Dryad or other) for increased findability and accessibility, and adding a complete data dictionary, that could be provided as an additional sheet in the excel file or as a separate file in the online repository.
- For the sake of transparency and reproducibility, I also suggest giving more information about the density models used (especially REM, but also SERC and other models). This could include an additional table with all parameters used in the REM model (species-specific activity levels, movement speed, detection area, etc.), and/or the raw data and codes used to calibrate and run the models.

Methods lack some important information for interpretation and reproducibility:

- Supplementary data 1: the two last sheets ('Energetics_byMassClass' and 'NPP') are empty.
- Please provide a description of ESWI (L 192; Figure 2) in the methods: how it was calculated, ideally with an equation,
- Field sites (L 259): Please provide the years of logging (subsequent rotations) in order to assess the time difference to the monitoring years.
- Field sites (L. 274): what is the area of unlogged plots? How many census years were included? There is no mention of oil palm plantation plots: is this an omission, or, if not, how was the NPP of the oil palm plantations calculated?
- L 307: because of the absence of data on bats in oil palm plantations, their abundance is estimated to zero in these types of forests (in the supplementary data 1). Do you have any reason to believe there are no bats in oil palm plantations, or if not, how do you address this potential bias?
- L. 342: Could you cite the relevant literature from which the species densities were compared?
- L 378: Could you provide a source that mentions that these species have particularly mobile life histories? It would also be useful if you could briefly explain why this can create a bias. Please cite the literature used to correct the densities of these species. How are home ranges used in the model to estimate population densities? I'm sorry for the naive questions, but I think some justification would be helpful to other non-specialist readers.
- L 396: it wasn't clear to me what were the predicted variables in the model (guild energetics?), nor where the results were reported.
- L 413, 415, 421--423 – I didn't find any justification for the uncertainty values chosen in this section, e.g. why is mammals' body mass more variable (20%) than birds' (10%)? Where do all these uncertainty estimates come from?

Appropriate use of statistics and treatment of uncertainties

Some numbers reported in the main text (e.g. L 97, 152, 157) and in the figures lack uncertainty

estimates. Is this to make the text more readable? If so, please provide this information elsewhere, or cite where it can be found (if in Supplementary Table 1 for example).

Error bars in Figures 1 and 3 should be described in the legends.

L 118: The acronym CI should be spelled out and the range (95%?) reported.

L 195-197: the name of the test performed should be indicated, and the name of the statistic provided should be spelled out.

Conclusions: robustness, validity, reliability

The conclusions seem to be supported by the data and by previous studies in the same study area. They could however be different in other regions with different species communities or under different disturbance regimes (e.g. with higher hunting pressure, as discussed in the main text).

Suggested improvements: experiments, data for possible revision

See suggestions in section C for additional data. I don't think any additional experiments are needed.

Clarity and context: lucidity of abstract/summary, appropriateness of abstract, introduction and conclusions

The text is well-written, the abstract provides all the key information. The introduction provides essential context for this study, and the conclusions are concise and supported by the data analysis. I think, however, that the introduction could provide a little more context with other work on similar topics, in terms of the effect of land use/forest type on species diversity and abundance, and on ecosystem processes. For example, citing other studies that have examined the effect of different types and intensities of logging on wildlife diversity (e.g. Burivalova et al. 2014; Bicknell et al. 2014; full citations at end) or ecosystem functions (Schleuning et al. 2017).

Other minor comments

L18: missing parenthesis

L 93 'biomass mass' is redundant

L 114 – please provide a reference or data supporting this statement

L 410 – citation not in Nature style and not included in the References

Fig 1: (f) % NPP : y-axis not the right unit (should be % instead of $\text{kJ m}^{-2} \text{ year}^{-1}$)

Figure 1 – the colour legend doesn't correspond to the panel letters (a and d-g should respectively be e and c, d, f, g)

Figure S3: the legend should describe the colours used and their acronyms.

Figure S4: the panels mentioned in the legend don't correspond to the figure

References:

Bicknell, J. E., Struebig, M. J., Edwards, D. P., & Davies, Z. G. (2014). Improved timber harvest techniques maintain biodiversity in tropical forests. *Current Biology*, 24(23), R1119–R1120. <https://doi.org/10.1016/j.cub.2014.10.067>

Burivalova, Z., et al. (2014). Thresholds of Logging Intensity to Maintain Tropical Forest Biodiversity. *Current Biology*, 24(16), 1–6. <https://doi.org/10.1016/j.cub.2014.06.065>

Schleuning, M., et al. (2011). Forest fragmentation and selective logging have inconsistent effects on multiple animal-mediated ecosystem processes in a tropical forest. *PLoS ONE*, 6(11). <https://doi.org/10.1371/journal.pone.0027785>

Author Rebuttals to Initial Comments:

Response to Reviewers

We thank the referees for their thoughtful and considered reviews.

Below we go through each of the comments in turn and give our responses *in green italics*.

Referee #1

Positive

Originality and significance:

Their work addresses novel questions. Although previous work has compared mammalian and bird communities across gradients of land-use change including old growth forests, logged forests, and/or oil palm plantations (e.g. Wearn et al. 2017 - Biol. Conserv.; Costantini et al. 2016 - Biol. Conserv.; Wearn et al. 2016 – Ecol. App., Wearn et al. 2018 - J. App. Ecol), I am not aware of any previous research adopting an energetic approach to compare different forest classes in tropical regions. I consider this novel aspect, quality of the datasets and quality of the presentation as key strengths of their research.

We thank the referee for recognising and highlighting the novelty of our approach (in particular the novelty of applying an energetic lens) and the quality of our datasets and presentation. Beyond this particular study we think we have demonstrated how such an energetic approach is more widely applicable for analysis of ecosystem change.

Data, methodology & appropriate use of statistics and treatment of uncertainties:

There are, however, some methodological points that may need some clarification. For example:

1 - Would differences in sampling effort across their land-use classes affect their results?

Sampling effort in logged forests was 1.7 higher than in old growth forests and 2.6 higher than in oil palm plantations.

Our analysis is based on species densities (and in particular densities of common species) rather than detection of rare species (which a biodiversity-focused analysis would have). Rare or hard-to detect mammal or bird species matter very little for energy flow and, as long as sampling effort is substantive, we should not expect differences in sampling effort to affect our estimates of densities. Moreover, the analyses account for imperfect detection, e.g., for the camera trap analyses, the number of camera trap nights are in the REM model, and so sampling effort is explicitly controlled for. For estimates of density of common species, sampling effort affects precision only and does not introduce a bias in absolute numbers (as it would do for species counts).

2 - Were their logged study sites distributed across a gradient of logging intensities and what type of logging operations were conducted there?

Reduced-impact logging and conventional logging result in distinct impacts for tropical biodiversity (e.g. Bicknell et al. 2014 – Current Biology). Moreover, logging intensities vary across areas within a concession and oversampling low-impacted regions could bring extra confounding effects that may affect their biodiversity assessments and energy flow estimations.

Logging intensity was high throughout the logged forest landscape with all sites experiencing between two and four rounds of rotation. All logged sites were conventionally logged - none experienced Reduced Impact Logging or anything near it. The SAFE sampling design (the basis of the survey design for mammals

and birds) was random with respect to logging intensity but ended up having good representation of all logging intensities in the landscape.

To assess how the various sampling points are distributed across the intensity of logging, we employ the biomass estimates at 30 m resolution for the landscape derived from airborne LiDAR by Asner et al (2018) as a proxy for logging intensity (i.e., timber removed/remaining). This is shown in the new Figure S1, repeated below. At this resolution there are a broad range of biomass values in both old growth and logged forests, but the mean values are clearly distinguished. The sampling points for vegetation primary productivity, bats, birds and terrestrial mammals (Figure S1b) span this gradient well.

Figure S1: distribution of sampling locations across a gradient of logging intensity, characterized using aboveground biomass (t/ha). We estimate biomass from a spatially-explicit surface of carbon density (30m resolution) derived from airborne Light Detection and Ranging (LiDAR) data (See (Asner et al. 2018) for full sampling details) and convert carbon to biomass using a conversion factor of 0.47 (Martin and Thomas, 2011). To provide a representative sample of local habitat conditions, biomass was extracted as mean values from 100m radii buffers around each sampling point.

Figure S1a

Figure S1b

3 - How is the landscape context of their work and could time since logging be driving their outcomes?

A higher amount of old growth forest [source populations] in the landscape may explain the lack of differences in bird diversity. Alternatively, if old growth forests have been impacted by other disturbances (e.g. hunting, fires, illegal logging), it could be that shifted baselines would influence energetic values observed in this forest class. On the other hand, the 40-46 years interval between the first round of logging (the 1970s) and their surveys (between 2010-2016) could mean that results reflect the reality of logged forests after many years of recovery -- and this [time since logging operations, post-logging interventions and inactive concessions – e.g. Burivalova et al. 2020 – Biol. Conserv; Cerullo et al. 2019 – For. Ecol. Manag.] is not recognised in their caveats.

We have now added a map (Figure 1) and accompanying text that better explain the sampling design and landscape layout.

Old growth forest as source regions

There is a very low amount of old growth forest in the immediate vicinity of logged forest landscape (Figure 1). The old growth forests in the study are either too small (VJR, area = 22 km²) or too distant (Maliau, distance = 65 km, Danum Valley, distance = 50 km) to act as substantial source populations for the SAFE logged forest landscape (area = 72 km²). However, the SAFE landscape is on the edge of a larger area of more moderately logged forest to the north, and we now mention this in our caveats (that more isolated fragments of logged forests might be more defaunated than those studied here: lines 506-507). The

possibility of remnant old growth forests being source regions of course does not explain away our key observation that logged forests have much higher abundances and energetics than old growth forests.

Shifted baselines

While there might be some declines in faunal abundances in VJR due to its position as a large fragment between logged and oil palm landscapes, the Maliau, Danum and Sepilok study sites are well-protected large areas, experience very low hunting pressure and are unlikely to have experienced any substantive decline in energetically dominant fauna because of anthropogenic pressure (Wearn et al 2017). These sites are the focus of intensive research and in the literature available to date there is no indication of the old growth forests being unusually depauperate compared to other old growth forests in Borneo. Indeed, many studies point out how intact and abundant the fauna appear to be because of low hunting pressure (e.g. Wearn et al 2019). We have added this point into our discussions in the Supplementary Material (lines 684-694).

Please also see our detailed response and analysis below to Reviewer 3 about the landscape autocorrelation or regional differences

Recovery in logged forests over many decades

These forests have been logged a number of times, with the most recent logging in the mid-2000s (Struebig et al 2013). As shown from the structure and biomass (Figure S1, and Figure S8 repeated below), these are very far from structural recovery). As suggested, we now discuss this point in the Supplementary Material (lines 696-707).

Figure R1: LiDAR measurements of forest structure in some of our old growth plots (Maliau Basin) and heavily logged plots (SAFE), showing how the logged forest are heavily structurally degraded and far from structural recovery. From (Milodowski et al. 2021).

Conclusions:

Despite my concerns highlighted above (and below in the minor comments), the study and approaches seem robust and valid. Further suggestions and clarifications on my concerns are provided below.

Minor comments:

L14: Declines in logged forests have been observed even for non-specialist species (depending on logging intensities and approach adopted – e.g. Burivalova et al. 2014 – Current Biology, França et al. 2017 – Biol. Conservation). Would be interesting if the authors could recognise that somewhere in the text.

We have modified the text (lines 45-46) to incorporate this point.

L27-28: Is there any available data and/or references that could support this claim of the importance of invertebrates as a substantial constrain on vegetation growth rates?

We are not aware of any literature demonstrating this point. This is a point of speculation based on the high rates of invertebrate herbivory implied by our calculation. Despite being speculative we believe it is an intriguing point worth mentioning: if the higher end of the range of NPP appropriation estimates are true it is hard to envisage such levels of herbivory not having an impact on growth rates. We have rephrased this sentence to highlight this is speculation (146-152).

L37-38: Low ecological value compared to which baseline?

I would restructure the second part of this sentence. Although studies have shown that human-modified forests have a higher ecological value when compared with deforested areas for agriculture (without claiming that human-modified forests/ecosystems are suitable for conversion to other land uses) there is a large body of research showing the importance of pristine /undisturbed ecosystems for maintaining biodiversity (high ecological value) when compared with human-modified ecosystems (low ecological value) – e.g. Gibson et al. 2011 – Nature; Barlow et al. 2016 - Nature).

We have added "compared to old growth forests" to clarify the baseline in this sentence (lines 38-39). We make the point in several places in the manuscript (e.g., lines 33-35, 263-266) that old growth forests have high biodiversity value. However, a key message of our study is that in terms of ecosystem energetics (and associated ecosystem function), logged forests have equivalent or even higher ecological value than old growth forests, something that has not been demonstrated before.

L42: Not only old growth specialist species... overall biodiversity loss have been documented in logged forests at the community level and associated ecological functions for distinct taxa and multiple (forest specialist and non-specialist species) – e.g. Burivalova et al. 2014 – Current Biology; Burivalova et al. 2015 – Proc. R. Soc. B, França et al. 2017 – Biol. Conservation.

We have modified the text (lines 45-46) to incorporate this point.

L58-61: Any references on how the energetic pathways can be linked to ecosystem processes and trophic factors?

We have added references to the classic books by Odum (1973) and Reagan and Waide (1996), which alluded to this point based on a detailed energetics study in Puerto Rico in the 1970s (lines 58-64).

L78: An amazing sampling effort!

Thank you!

L82: What “34 site-years” mean? I recommend some clarification on this.

This is the summation of site (i.e. plot) x number of years monitored. We now clarify this in the main text (line 85).

L88-89: Would the maintenance of bird diversity be related to (a) shifted baselines of old growth forests and/or (b) landscape context (e.g. amount of old growth forest in the region)? Have you accounted for these (a,b) during your models/uncertainty calculations?

See our response above on the landscape context of the study area. Although meta-analysis for Borneo show bird abundances generally decline in logged forest (Costantini et al., 2016), most of the studies included within that appraisal did not control for imperfect detection - this is a key asset of our study. In this revised version, avian diversity estimates were derived from detection-error-corrected estimates of the true presence absence matrix (known as the Z matrix). When summed, the Z matrix provides an estimate of species richness corrected for imperfect detection. We took the total median occupancy probability for each species at each site as an estimate of overall species richness. Previously, richness was calculated via the sum of all species with occupancy probabilities greater than zero. This does decrease the estimated species richness in our estimates and introduce more variation across habitat types (Figure 2b). However, it still shows higher species richness in the logged forest compared to the old growth forest.

L142-147: Any references available for these claims?

See our response above (to L27-28) on this same point.

L194 and L263-266: Are there any information on the (a) logging intensity and (b) time since logging at the study site level?

Time since logging is a key factor driving ecological patterns in logged forests. When were the subsequent rotations?

Logging intensity is not uniform across concessions, but is also a key driver of ecological patterns in tropical forests (e.g. Burivalova et al. 2014 – Current Biology, França et al. 2017 – Biol. Conservation). Do the authors have access to logging intensity values for study sites?

Assessing data on these (a & b) is key when contrasting these two forest classes and lack of differences could be a result of low logging intensity and/or shifted baselines. There is information provided in L263-266, but these seem more like averaged values across the sites. Yet, the ground reality of logging concessions sometimes means that some sites /regions in the concession have a higher volume of timber extraction compared to others sites/regions. Thus, knowing whether surveyed sites have been distributed across a gradient of logging intensities would be helpful to understand if there were any biases towards low- or highly-impacted forests.

We have expanded our site description (lines 275-293) and added in a new Figure 1 (a map of the sampling sites). We have a broad logging history for the landscape but not for specific plots. The LiDAR-derived biomass data (Figure S1, repeated above) for the landscape give an indication of where our sample plots fit within the study landscape. Current biomass/canopy height can be taken as a measure of past logging intensity. As the referee points out, there is large spatial heterogeneity in logging intensity, but our sites span the range of past logging intensity quite well. Overall, logging intensity has been higher at SAFE than at most sites considered globally (cf (Burivalova, Sekercioglu, and Koh 2014)). Therefore, it is not an explanation for the relatively high 'intactness' of the mammal/bird communities. See also our response to a similar point above.

L267-268: I would expect logging roads to facilitate access to hunters with further impacts particularly for large-bodied terrestrial birds.

Levels of hunting at this site are very low compared to other sites in Borneo (See (Wearn et al. 2017), Supp 1.2) for detailed discussion of this. We see this study as a high degradation-low defaunation site to examine primarily the effects of structural degradation alone on ecological energetics, without defaunation as a confounding factor. We now emphasise this point in our conclusions, that this study represents a best case study if extensive hunting can be avoided in logged forests (lines 270-272). Please see also our new sensitivity analysis of the effects of hunting (Figure S4, discussed further below), which draws on this point.

L292-293: Sampling effort in logged forests was 1.7 higher than in old growth forests and 2.6 higher than in oil palm plantations. Would that be a factor driving /affecting the study outcomes?
I mean, higher sampling efforts may result in more detections /density estimations.

Please see our response to the same point above. Sampling effort is unlikely to have any significant effect on our abundance-focussed metrics.

L306-307: Would that affect their comparisons/results?

This comment refers to whether avoiding rainy days would affect bird abundance estimates. Avoiding days of rain is best practice when conducting point counts, because it helps to control for the reduced detection probabilities associated with lower levels of bird song. Only 3 days of point counts were lost to rain during the entire sampling campaign.

L322-323: I am not a specialist in camera trap work and was wondering if the REM approach accounts for the same individual passing through the (same or different) cameras in a given territory / region and how this could influence density estimations. Would it be possible to provide some clarification on this?

REM does accommodate individuals passing through a camera multiple times (which likely happens often), or indeed multiple cameras. We make no attempt to identify individual animals in the REM approach, given the difficulties of doing so. Instead, REM density estimation is based on a gas equation, which has been modified to account for the specific way in which camera traps and animals 'interact' (Rowcliffe et al. 2008). The gas particles in this case are the (static) camera traps and the (moving) animals, and contact rate is a function of the density of particles (the number of camera traps is known, but number/density of animals is not and is the parameter we are solving for), the speed of the particles (animal speed was estimated from the camera trap data; the camera traps are obviously stationary), and the size of the camera trap detection zone (a larger detection zone would result in more 'contacts'; this too has been estimated from the camera trap data).

In practice, smaller species, and herbivorous species, typically have small home-ranges, and move slowly, and therefore contact a small number of cameras relatively infrequently (e.g. consider a muntjac or mousedeer). On the other hand, carnivores (e.g. clouded leopard) move quickly and contact a large number of camera traps. These aspects of the ecology of different species are controlled for in the REM approach (e.g. so that a relatively large number of detections of clouded leopard does not mean that the density is estimated high; instead the model 'knows' that it is a relatively small number of individuals that are moving far and wide).

L341-342: Any references for this?

Presuming this request refers to the R package secr, two references are now provided (Borchers and Efford 2008; Royle, Andrew Royle, and Young 2008).

L363-364: If each site received 4 visits, would that mean that the maximum detection would be four? Or would multiple individuals be accounted in a single sampling occasion? Clarifying this would be very helpful.

In the case of the bird sampling, multiple individuals within a sampling occasion would not be accounted for, a species is simply present or absent within a sampling occasion, thus the maximum number of detections for any given site would equal the number of sampling occasions for that site.

L400-410: It would be interesting if uncertainty calculation accounted for differences in sampling efforts across the land-use classes that were compared.

Sampling effort is incorporated via the population density uncertainty, as those values are drawn directly from the bootstrapped runs of population density models (mammals) or sampling (birds) for each species and habitat, which reflect the sampling effort.

L442: There is an extra parenthesis here.

Corrected.

Referee #2 (Remarks to the Author):

This is a highly innovative and very detailed analysis of shifts in ecosystem energetics in tropical forests after logging. It is unique and innovative because it is the first analysis of shifts in energy flows due to logging, and provides a first and new assessment of the functioning of logged tropical forests. The study provides a very detailed account of consumption pathways across trophic levels, making use of sophisticated empirical data on changes in primary productivity and abundance of animal groups between logged and unlogged forests. By comparing plots in a logged and an unlogged forest area in Borneo, the authors show that NPP and animal abundance is higher in logged forests, and so is the energy intake by bird and mammals.

The study changes the way we perceive logged tropical forests and their ecology: from ecosystems that are functionally and structurally "degraded" and have lost species and carbon; to ecosystems that – in spite of structural alteration – possess a much higher productivity and faunal abundance, exhibit enhanced energetic pathways, and resilient energy flows. Given that a large share and a large area (400

mln ha) of tropical forests are used for timber production, the paper provides a highly relevant message on an “alternative” way to see these forests: as fully functional ecosystems with a potentially high level of resilience.

We thank the reviewer for identifying and highlighting the novelty and value of this study.

I have three concerns about the study:

1. Dietary shifts. The calculation of changes in energy flows between forest types is solely based on differences in abundances (which – to my knowledge – are estimated appropriately), assuming that logging does not cause shifts in diet or feeding activity of bird or mammal species. I wonder to what extent results are robust to possible changes in feeding behaviour and trophic level (diet). Although this is not my field of expertise, I found an example of a study documenting strong dietary shifts (as much as one trophic level) of birds in logged vs undisturbed forests (Hamer et al 2015). The shift in logged forests to a higher abundance of pioneer tree species, understorey shrubs/herbs and thus palatable leaves may increase insect herbivory (which unfortunately was not included in the measurements) and hence shift diet of birds and mammal species. While I acknowledge and appreciate that shifts in diet were to some extent treated in the uncertainty analyses, I do not think that this analysis allows to evaluate the effects of a pervasive shift in diet. I therefore suggest the authors conduct a robustness test to evaluate whether conclusions are robust to observed (see Hamer et al 2018) or expected dietary shifts due to logging.

We do include a random uncertainty in our assignment of diet preferences but do not include a systematic uncertainty as there is little evidence of a shift in diet types across the land use types. As we focus on the energetic uptake through various food pathways, the only way that changes in diet affect our calculations is through variation in assimilation efficiency. The assimilation efficiency is very similar among most food types (around 70-90%; see Table S3), with the notable exception being leaves, which are less nutritious and palatable (assimilation efficiency around 30-50% in mammals, apparently higher in birds where they are a minimal part of diets). Hence the only shift in dietary type that would have a noticeable impact on our calculations would be one from leaves to other foodstuffs. There is very unlikely to be a major shift in diet from leaves to other food types because of constraints of feeding and gut morphology. There is more evidence of dietary shift within a feeding guild, e.g. from feeding on herbivorous arthropods to predators (e.g. spiders). We directly consider this shift in trophic level of insectivorous feeding (as reported by (Edwards et al. 2013)) in our calculations of indirect energy consumption. The more of an upward shift in trophic feeding level there is, the greater the indirect consumption of NPP via the arthropod pathway (but the direct consumption of NPP remains largely unaltered).

To test the sensitivity to dietary shift we ran an additional model with a 30% shift towards arthropods for the mixed feeders. If we assume a 30% shift towards leaves at the expense of seeds, nuts and fruits for mixed herbivores in logged forest, this results in only a small (2.3%) increase in energy consumption compared with no shift in their diet.

We have added discussion of this point in the Supplementary Material (lines 709-727) and refer to it in our caveats (lines 496-499).

2. Oil palm. I do not see the need or added value of including the results of the oil palm plantation in this manuscript. The text is now framed in relation to selective logging in tropical forests, the main messages are about logging effects on energy flows, and the main innovation is in the quantification of logging effects on energy flows. In my view, there is no reason to include the oil palm results, also not

as a contrast with other forest types. Including oil palm may only divert attention from the main messages on logging effects. I therefore recommend leaving out the oil palm results.

We agree that the oil palm results are not a core part of our results, but feel they do add value in showing how the amplified energetics do reverse on conversion of the logged forests to oil palm (something that we argue a wrong lens of degradation can encourage). It adds a suitable contrast to the logged and old growth forests, and a point of reference to what logged forests are converted to if they are considered degraded forests..

3. Representativeness. This study was done in one region in Malaysia, with high logging intensity (compared to Neotropical and African logging intensities) and low hunting pressure (compared to most logged forests, probably). I missed a discussion on the representativeness of this study for other logged tropical forests, both within and outside SE Asia. To what extent are similar shifts in NPP, faunal abundance and energy flows expected at lower logging intensity? The reported NPP shifts can be compared to similar studies in other regions (GEM studies in Brazil; other studies on woody productivity in logged forests), to provide context. To what extent are the reported increase in faunal abundance expected at higher (and more representative) hunting intensity? The study site can be easily put in context using defaunation indices. Finally, the particularly low hunting pressure in the study region also requires some discussion of the expected shifts in ecosystem energetics under higher hunting intensity.

In terms of NPP, the old growth plots in Borneo are in the range of plots measured in Amazonia (Malhi et al. 2015) and in Africa (Moore et al. 2018), as well as old growth forests at Lambir in Sarawak (Riutta et al. 2018). Forests in Borneo do seem to allocate more NPP to stem growth (Riutta et al. 2018), leading to generally higher rates on wood production in Borneo than in other regions

Comparison with other sites

We have not yet conducted studies of shifts in NPP in response to logging in other regions and, to our knowledge, no such published studies exist. In the case of these Borneo plots, we demonstrated that relatively intact stands in logged forest showed an increase in NPP, probably in response to more light availability, but this was offset by canopy loss in heavily degraded patches (logging platforms etc.), leading to only modest change in NPP overall (Riutta et al. 2018). Hence it is possible that more lightly logged forest, with fewer heavily degraded patches, to have even higher NPP. These factors (light availability vs heavy degradation) are likely to be ubiquitous. However, Bornean forests tend to be more heavily logged than others because of the high density of timber species (especially dipterocarps).

Schleuning et al (2009) examined a range of ecosystem processes (pollination, seed dispersal, seed predation, decomposition, army-ant raiding and antbird predation

in intact and logged forests in Kenya and found that all measured ecosystem processes were amplified in the logged forests, a finding consistent with our study. We have included a reference to this in our text

We could not identify many other studies that quantify species abundances after logging (as opposed to diversity), and none that looked across all bird and mammal taxa (rather than selected taxa). However, most tended to support patterns of either no change or increased abundance in logged forests. The best evidence we could find was also from Borneo: (Cleary et al. 2007) showed that a number of bird generalist species increased in abundance in logged forest. The response of mammals seems quite variable, depending probably on the intensity of logging and the degree of hunting present. (Brodie, Giordano, and Ambu 2015) found that large mammals generally increased in abundance in logged forests in Borneo, but

carnivores decreased. (Granados et al. 2016) compared lightly logged to old growth forest in Danum Valley and did not find significant changes in abundance for the six species they studied. In lightly logged forests in Eastern Amazonia, (Almeida-Maués et al. 2022) found "no discernible difference in the abundance of medium-large mammals between primary and reduced impact logged forest".

It seems the energetics effects for birds are amplified even more heavily logged forest compared to lightly logged forest, but not so for mammals (Figure S5).

Effects of hunting intensity

The reviewer's comment about the effects of hunting intensity stimulated a further analysis which proved an interesting and valuable addition to the manuscript.

As a sensitivity analysis of the effects of defaunation, we explored the possible energetic consequences of 50% reduction of species affected by targeted hunting by indiscriminate hunting or by both, assuming no feedbacks on the densities of other species or on NPP (Figure S4, repeated below). Targeted hunted species include commercially valuable birds and gun-hunted mammals (bearded pig, ungulates, banteng and mammals with medicinal value). Indiscriminately hunted species include birds and mammals likely to be trapped with nets and snares. For the list of species in each category, see Supplementary Table 1. This analysis shows extensive targeted and indiscriminate hunting bring bird energetics to levels close to (but still above) old growth forests. For mammals, however, even extensively hunted logged forests seem to maintain higher energetic flows than old growth forests.

Hence our conclusion is that defaunation can be important, but only extensive defaunation "offsets" the enhanced energetics in logged forest. This highlights the importance of avoiding intensified hunting pressure in logged forests, which acts against the amplified ecosystems energetics. This quantification provided a valuable new insight, and we have added a few lines of discussion of this point in the main text (lines 218-229) and included this new figure in Supplementary Material (Figure S4).

Fig. S4: Body mass of birds (a) and mammals (b) and energetic food intake of birds (c) and mammals (d) in old growth forest (OG, dark grey, no hunting) and in logged forest (light grey) under four different hunting scenarios: observed low hunting pressure (baseline), and simulated 50% reduction in population density of targeted hunted species, indiscriminately hunted species and both targeted and indiscriminately hunted species. Targeted hunted species include commercially valuable birds and gun-hunted mammals (bearded pig, ungulates, banteng and mammals with medicinal value). Indiscriminately hunted species include birds and mammals likely to be trapped with nets and snares. For the list of species in each category, see Supplementary Data Table 1. Note that this is not an exhaustive analysis of the hunting pressure in the study area, but an illustrative estimate of the potential impact of hunting on trophic energetics.

Other comments:

- Abstract. At present, the abstract (second sentence) does not convey the main gap in knowledge that is addressed in the study. Also, I thought that the expression that a "lens shines a new light" was a bit strange: lenses allow light to pass, but do not actively shine... I certainly do like the term "energetics lens", as this is what the study does: helping us to view logged forests differently. Third, I think it is important to stress that this study was conducted in an area with low hunting pressure, implying that the reported enhanced energetics pathways in logged forests may be jeopardised by high levels of hunting.

We have modified the second sentence so the abstract now starts "Logged forests are often characterised as degraded ecosystems, due to their altered structure, loss of biomass and declines in intact-forest specialist species. However, whether this also corresponds to a degradation in ecosystem functions is less clear: shifts in the strength and resilience of key ecosystem processes in large suites of species have rarely been assessed in an ecologically integrated and quantitative framework."

We have changed "shine a new light" to "gain new insight"

As discussed above, we have conducted a quantitative analysis of the effects of hunting and discuss this more in this new version ((lines 218-229 and Figure S4). We have not added this point to the abstract because of word length constraints and because it is nuanced - our analysis suggests that logged forests experiencing high levels of hunting experience reduced bird and mammal energy flows, but to at levels below those of faunally intact old growth forests.

- Figure 1. I found the layout of this Figure to be somewhat unorganized and think the quality can be improved (to the standard of Figure 2). It would be logical to start (upper panels) with the NPP and probably with animal densities, and then move down to energy flows. In the present Figure the NPP is somewhat hidden in the middle. Also, I do not get why energetic intake is expressed in kg biomass, while all other productivity measures are expressed as kJ. (One of the claims the authors make is that ecosystem energetic studies allow comparisons using a common metric (energy, kJ), but here this does not seem to be the case.) Finally, it is unclear what the whiskers mean.

We agree the figure was not optimal, so provide a new and much improved figure (now Figure 2). The units of energetic intake were incorrect in the previous version.

- The results of the uncertainty analyses did not make it to the main text, and I would argue that this is a miss. These are important analyses, skilfully conducted and too important to be dealt with in the Methods section alone.

We have added some more text on the uncertainty analysis to the Methods and clarified what was there previously (lines 442-487).

- I could not find results of the mixed effects models that are mentioned in the Methods section.

We have added text on the mixed effects models to the Methods (lines 436-440).

Referee #3 (Remarks to the Author):

This manuscript addresses differences in energetic pathways through plant-bird-mammal communities across undisturbed and logged tropical forests and palm plantations. Data on NPP and densities of birds and mammals on a Borneo field site are used to estimate consumption rates. The study is coming to the conclusion that "Far from being degraded ecosystems, even heavily logged forests can therefore be vibrant and diverse ecosystems with enhanced levels of ecological function." Overall, I found the topic of the study intriguing and important, and the manuscript is very well written. However, I have concerns that the methods applied can be used to support the conclusions. Despite this critical feedback on the methods and the presentation of the results, I hope that my points can help to strengthen the manuscript.

We thank the referee for finding the work intriguing and important, and address their feedback below.

1) While the conclusions are drawn for degraded ecosystems in general, I have the impression from the methods that just one logged forest has been studied. I might be wrong here but the methods are not very explicit about the sampling site(s?), and such information would be crucially important to judge the level of generality that can be achieved. I certainly understand the constraints of field sampling but if my understanding is correct that this is just one area of logged forest compared to another area of pristine forest, I would be more careful with the conclusions as the results could also just represent one specific case.

We agree that more detailed description of the sampling was needed, so have included a more detailed description and a new Figure 1 that maps out the sampling design (also see our responses to other referee comments above).

A broad landscape of logged forests was sampled (Figure 1), and four old growth forests (two for both birds and mammal, two more for birds only). We have added a caveat about general applicability, but the broad ecological explanations we explore suggest these insights are general and not site-specific. Our study landscape is more intensively logged than most non-Asian forests, but also less hunted than many logged forest regions. We spell out these caveats more clearly in the main text (e.g., lines 218-220), and also now highlight in our conclusions that this study represents a low-hunting best-case scenario, showing the potential for greatly amplified energetics in logged forests if heavy defaunation can be avoided (lines 270-272).

2) I am missing a section describing the statistical approaches. How did the authors compare the energy pathways across the three forest types?

We have now added text to the Methods explaining in more detail the statistical methods (lines 436-487) and how we compared energy pathways across the habitat types.

3) What is represented by the error bars in the figures (e.g., Fig. 1)?

For all figures, the error bars denote 95% confidence intervals, derived from 1000 Monte Carlo simulation estimates, incorporating uncertainties from multiple sources. These are now explained in more detail in the figure captions.

4) Have the authors considered spatial autocorrelation in their statistical analyses? The description of the study sites is a bit vague but I am under the impression that the logged forest is at one site and the unlogged forests at another site. If this holds true the results are challenged by the alternative

hypothesis that spatial effects are driving the differences in densities and thereby energetics. The statistical analyses should test this hypothesis to show that it can be discarded.

Please see our discussion above about the landscape layout and our new Figure 1 that maps the layout. The logged forest sites are spread across the single broad landscape, the old growth forests are spread between a site adjacent to the logged forest landscape (VJR), and other sites that are spatially separated (Maliau for mammals, and Maliau, Danum and Sepilok for birds). Our sampling unit is ecosystem type, not individual sampling points.

If our results were driven by strong spatial contrast between sites (e.g., geological or environmental factors), we would expect strong differences between the local old growth site (VJR) and the more distant ones. Conversely, if local source-sink effects were affecting our proximal old growth site, we would also expect differences in energetics between local and distant old growth sites.

To examine differences between the various old growth forests we separately analysed the bird and mammal energetics for VJR, Maliau (birds and mammals) and Danum and Sepilok (birds only; see below, also now Fig S5 in the Supplementary Material). This shows relatively little difference in overall bird and mammal energetics between the various old growth sites compared with the logged forests, and the logged forest energetics are consistently higher. This strongly suggests that both geographical and source-sink effects have relatively little effect on our results, and the high energy flow in logged forests is driven primarily by the direct consequences of the logging. For birds, the consumption energetics were even stronger in the more heavily logged forests, but this was not the case for the mammals.

We have added this discussion point to the Supplemental Material (lines 729-748).

Figure S5: Body mass and species richness of birds (a) and mammals (b) and energetic food intake of birds (c) and mammals (d) across old growth forests (OG), logged forests and oil palm plantations (OP). Old growth forest data were analysed separately by four old growth sites for birds and two sites for mammals (see Fig 1 for map), and the logged forest data were split into twice logged and heavily logged areas. For mammals, only species studied using camera traps and harp traps were included (63%, 63% and 77% of mammal species, and 55%, 45% and 63% of total energetic food intake in OG, logged forest and OP, respectively). Error bars are 95% confidence intervals derived from 1000 simulated estimates incorporating uncertainty in body mass, population density, the daily energy expenditure equation, assimilation efficiency of the different food types and composition of the diet of each species.

5) From the methods description, it appears that the NPP measurements, the mammal and bird surveys have been carried out at different locations. Again, the description is not detailed enough here to allow a thorough evaluation but NPP has been assessed at 5 (logged forest) and 4 (old growth forest) sites. Camera traps have been set up at 882 locations. Terrestrial mammals have been caught at 1488 locations. Bats have been captured at 42 sampling locations. Avian points counts have been carried out at 356 locations. This description suggests that the data have been obtained at different locations, but there is no description how the data have been grouped or combined to allow for energetic analyses. In addition, the samples have been obtained in different years making it even harder to compare them.

The description of the sampling design needs to be given with much more detail on the sampling sites and how the data have been pooled. Overall, however, I am afraid that I am not convinced that such data can be used to estimate energy flows through pathways.

We recognise we did not adequately describe the spatial and temporal sampling in the original manuscript, and have added a detailed description in the revised manuscript, as well as a new Figure 1 laying out the sampling design (see response above).

Overall, our measurements span the period 2011-2017. Vegetation productivity measurements span this entire period, and mammal and bird measurements were conducted at various times with this period (2011-2017 for medium/large mammals, 2011-2014 for small mammals, 2011-2012 for volant mammals, 2014-2016 for birds). These details are included in the Methods. We regard our analysis as providing an integrated view of these forest ecosystems over this half-decade. This would only be problematic if there were large variations in the composition of the community across space (within a habitat type) and time. We address the spatial variability across habitat types through our dense sampling strategy to ensure we result in a reasonable average for each sample type, which is now laid out in detail in Figure 1. As for temporal variation, there is little evidence that the aggregate density patterns of most long-lived communities are fluctuating strongly from year-to-year, but even if they were our analyses are based on average densities calculated over multiple years. The most likely group with annual variation is the small mammals ((Chapman et al. 2018), for which we base our estimates on four years of data. We see no evidence why these data cannot be pooled to estimate aggregate energy flows through pathways over this half-decade of focussed study.

6) I am skeptical that the estimates of energy demands via allometric equations can be used to predict energy flows. They are certainly an important component as they constrain the metabolic rates of the animals but the real-world demand is also affected by environmental factors such as the ambient temperature and interactions with other species. At least, a model predicting energy flows should include effects of temperature (along with allometries) and also the consumption loss rates of the populations. The latter point is based on the pattern that populations can be driven to low densities by predation despite high production rates (and thereby high rates of consumption and energy flow). A model such as the one used in this study ignores this possibility and is thereby too simplistic.

There are two points the referee makes here.

Do we need to account for temperature?

All our sites are all in a very similar tropical climate, and hence the major factor driving temperature differences between sites will be microclimatic effects caused by changes in forest structure. Previous studies in the same landscape show that these effects are modest, with the logged forests having peak daytime ambient temperatures around 1 °C warmer than the old growth forest, and 24-hour mean temperature difference being less than 0.5 °C (Hardwick et al. 2015).

For endothermic birds and mammals in warm climates, the sensitivity of metabolism to temperature appears to be negligible or very modest, with body temperatures maintained in the range 36-40°C ((Brown et al. 2004). A recent review (Portugal et al. 2016) finds that the relationships between the different components of wild mammal and bird daily energy budgets and environmental factors, such as ambient temperature, do not follow universal rules.

As a sensitivity test, if we assume a Q_{10} of 2.0 for DEE (i.e., a doubling of metabolic rate for a 10 °C warming) for the observed peak 1°C difference in local microclimate between old growth and logged forests, we would expect the bird and animal metabolism to be 7% higher in the logged forest than if there were no microclimate difference. For a 24-hour mean temperature difference of 0.5 °C, the increase would be only 3.5%. We do not include this calculation given the uncertainties involved, but it shows that any temperature effects are likely to be small, and that our key results are conservative in this respect (i.e. the contrast between logged and old growth forests may be slightly greater than we show here if temperature differences were to be taken into account).

Is allometric scaling with body mass sufficient to estimate energetic uptake?

The second point the referee makes is whether a model based on body mass scaling and population size alone is sufficient. We would like to emphasise that allometric approaches to estimating energy requirements are well-established and widespread in the ecological literature (e.g. (de L. Brooke 2004), (Croll, Kudela, and Tershy 2006), (Nyffeler, Şekercioğlu, and Whelan 2018)). We would argue allometric scaling is sufficient. By estimating the population extent, we use field metabolic rates to calculate total energy uptake via food consumption. Our analysis focuses on the consumption of energy and therefore only needs to know population densities and estimate food requirements, not how and where energy is invested (e.g., in reproduction vs maintenance). There is inevitable variation across species according the behavioural peculiarities (our error analysis shows that uncertainty in allometric relationship is the largest contributor to uncertainty (Figure S6)), but when summing across tens of species we would expect such differences to average out: indeed, we would argue that when analysing rich multi-species communities, no other approach is possible.

Consider a quasi-steady state population of 100 birds. It could be a slow turnover population where 1 bird dies every 10 days and is replaced. Or a high turnover population where one bird dies every day. Ultimately, the amount of food consumed by the population is largely the same because on any given day there are still just 100 birds (there may be minor second-order effects as the age structure of the populations will be different in the two cases, with a younger mean population in the second example). But, to a first order, the food consumption energetics can be adequately described by the mean body size of the adults and the number of individuals in the population, as we do in our calculation. In a shift from intact to logged forests there may be behavioural changes in daily energy expenditure linked to habitat, competition etc., but again we would argue that these are minor effects. Our results are overwhelmingly driven by the higher densities of populations in most bird and mammal guilds in the logged forests - behavioural changes would have to be massive (more than halving in field metabolic rates) to compensate for the increased abundance.

7) In addition, I am also not convinced that a study based on plants, mammals and birds can be used to estimate the energy flows through a community. The vast majority of energy flows goes through invertebrates that dominate most communities in general and tropical communities in particular in not only their biomass but also their per unit biomass consumption rates. This point is troubling me as the study results are used for very broad conclusions such as "The total flow of energy flux through consumption is amplified across all energetic pathways by 147% in logged forest relative to old growth forest." This conclusion is drawn despite the statement "Overall, up to 50% of ecosystem net primary production may be supporting the bird and mammal communities in logged forests, predominantly via invertebrate pathways, and herbivory by invertebrates may be a substantial constraint on vegetation growth rates." I am afraid that I am really not convinced that broad statements on energy fluxes can be supported by a study on plants and vertebrates.

We have gone through the manuscript and have been careful to make sure we always refer to flow through the birds and mammals where appropriate. Even if we could say nothing more about energy flow in the ecosystem, we argue this would be an interesting and novel measure of ecological vitality. These energetic approaches are able to combine biodiversity and abundance data to assess key ecosystem functions.

*We fully agree that more ecosystem energy flow is through invertebrates and, indeed, emphasise this in our results (e.g. the indirect consumption in Figure 2f, g). However, **we are able to say something meaningful about invertebrates**. As we argue in the paper, the abundance of invertebrate-feeding mammals and (especially) birds does tell us something useful about invertebrate abundances. For example, a much higher population of insectivorous birds in the logged forest requires high consumption of invertebrates to sustain them, suggesting a great increase of invertebrate populations (if population size is food limited, as seems likely given that we estimate that invertebrates consume up to 50% of NPP in the logged forest).*

Our findings are supported by a previous canopy fogging study in this landscape (Ewers et al 2015), which found an increase in canopy invertebrates in the logged forest compared to old growth, and an increase in total invertebrate biomass, even though litter invertebrate biomass decreased. We now mention this study in the main text to support our argument (lines 146-147).

Referee #4 (Remarks to the Author):

Summary of the key results

This paper quantifies energetic pathways from trees to birds and mammals in three of the major ecosystem types in Sabah, Malaysia: old growth forests, selectively logged forests and oil palm plantations. The results show that bird diversity (in terms of species richness and consumption pathways) remains high across the landscape, while mammal diversity is more sensitive to land-use conversion, and declines sharply in oil palm plantations. Total bird and mammal biomass and energy consumption are highest in logged forests, contrary to the common perception that old growth forests have a higher abundance of fauna than logged forests disturbed by humans.

Originality and significance

In my opinion, the most significant result of this study is the high ecological functionality of selectively logged forests (under low hunting pressure), that support a high abundance and diversity of birds and mammals. Some of these conclusions were already supported by previous results which include some of the data presented in this study: for example, Wearn et al. (2017; cited in this manuscript) showed that mammal abundance was higher in logged forests than in old growth forests and oil palm plantations. The originality of this new study is the use of energetic pathways that allows an integrative ecosystem-wide approach including most of the food chain of these ecosystems, from primary plant productivity to top predators. It thus provides a good overview of how land use conversion affects ecological interactions in this tropical landscape. The results would therefore be of significance to a broad scientific community, from landscape ecologists to conservationists.

We thank the referee for highlighting the originality and significance of our study, in particular the integrative insights that an energetics approach allows.

Data & methodology: validity of approach, quality of data, quality of presentation

I have little knowledge in fauna surveys (camera traps, etc.) and the models to infer population densities from such data, so it would be good to have the opinion of someone with more expertise on this. However, all the methods used have all been published in previous articles cited in references, and I didn't detect any major methodological flaws.

We have provided further details and responses around the survey methods and models in response to the other referees (lines 295-487).

The figures presenting the data are easy to understand and well designed. I have some reservations about Figure 3, which presents data that is less important for the main message and is only quoted once in the main text. I therefore suggest moving Figure 3 to the supplementary file, but of course I leave this decision to the authors.

Upon reflection, we agree with the referee on this point, and have moved that figure to the Supplementary Material (now Figure S3), which also makes space for our new site map as Figure 1.

Some suggestions to make the data provided more in line with the 'FAIR' principles:

- I would suggest putting the supplementary data 1 in an online repository (Dryad or other) for increased findability and accessibility, and adding a complete data dictionary, that could be provided as an additional sheet in the excel file or as a separate file in the online repository.

We will put the Supplementary Data Table 1 onto Dryad if this paper is accepted for publication.

- For the sake of transparency and reproducibility, I also suggest giving more information about the density models used (especially REM, but also SERC and other models). This could include an additional table with all parameters used in the REM model (species-specific activity levels, movement speed, detection area, etc.), and/or the raw data and codes used to calibrate and run the models.

We have added a sheet in the Supplementary Data Table that gives the density estimates for each REM modelled species and the associated parameters such as activity, speed, detection radius and distance. We think the raw data and code for the REM analysis would be too much here, and will publish them in a separate manuscript (led by Wearn) dedicated to the REM analysis.

Methods lack some important information for interpretation and reproducibility:

- Supplementary data 1: the two last sheets ('Energetics_byMassClass' and 'NPP') are empty.

Sorry these two were missing. We have filled in the worksheets in this resubmission.

- Please provide a description of ESWI (L 192; Figure 2) in the methods: how it was calculated, ideally with an equation,

This has now been added to Methods (lines 427-434), along with an equation.

- Field sites (L 259): Please provide the years of logging (subsequent rotations) in order to assess the time difference to the monitoring years.

We have added these details in the site description (lines 278-283).

- Field sites (L. 274): what is the area of unlogged plots? How many census years were included? There is no mention of oil palm plantation plots: is this an omission, or, if not, how was the NPP of the oil palm plantations calculated?

We have now added the information below to the Methods to answer this question (lines 296-300).

"Net primary productivity was measured in five logged 1 ha plots in the SAFE Project area with varying intensity of logging (five years of data), and in four old growth forest 1 ha plots in the Maliau Basin (two plots, four years of data) and Danum Valley Conservation Areas (two plots, two years of data) (Riutta et al. 2018, 2021), and one 0.36 ha mature oil palm plot (two years of data), following the standardised protocols of the Global Ecosystems Monitoring (GEM) network (Malhi et al. 2021)."

We have also now described how oil palm NPP is calculated (lines 304-307).

"Oil palm plantation NPP estimates were based on palm censuses and allometry with height, monthly counts of flower bunches, fruit bunches and attached and pruned fronds combined with a one-off survey of their mass, and quarterly harvest of the root ingrowth cores."

- L 307: because of the absence of data on bats in oil palm plantations, their abundance is estimated to zero in these types of forests (in the supplementary data 1). Do you have any reason to believe there are no bats in oil palm plantations, or if not, how do you address this potential bias?

We acknowledge this is a mismatch in data but thought it better to include bat data for the forest sites which are the main focus of our paper, where we seek to provide as complete a description of energetics as possible. There are likely to be bats in the oil palm. However, as bat populations have a fairly minor influence on energetics in both forest ecosystems, their influence on the overall patterns remains small and would not be altered by any reasonable estimate of bat populations in the oil palm.

- L. 342: Could you cite the relevant literature from which the species densities were compared?

Please see response to next point below.

- L 378: Could you provide a source that mentions that these species have particularly mobile life histories? It would also be useful if you could briefly explain why this can create a bias. Please cite the literature used to correct the densities of these species. How are home ranges used in the model to estimate population densities? I'm sorry for the naive questions, but I think some justification would be helpful to other non-specialist readers.

We have added a shortened version of the explanation below to the Supplementary Material (lines 750-768) and also include an extra figure here (Figure R2).

We used modelled density estimates to generate territory size estimates (assuming non-overlapping territories) and assess the allometric relationship with species' weight. We compared this with the allometric relationships between territory-size estimates and species' average weights for 27 Asian bird species, including 7 hornbills we were able to find in the literature (Fig R2b). Territory size estimates in the literature were based on alternate approaches including telemetry ((Poonswad and Tsuji 1994)) nest monitoring ((Kinnaird and O'brien 2008)) group-following ((O'brien 2008)), territory spot-mapping (Raman, 2003) and distance-sampling (Gale and Thongaree 2006); Raman, 2003). The allometric relationships

between body mass and territory size followed (Schoener 1968) (1968) and were broadly similar between our modeled estimates and those in the literature (Figure R2 below). However, our initial density estimates for large-bodied hornbill species were considerable outliers from this pattern (Fig. 1c). Issues with adequately estimating densities for hornbills based on point count methodologies have been highlighted previously, particularly when encounters of fly-over birds are included (Marsden 1999) as is the case in our dataset. In order to preserve differences in the relative abundance between hornbill species in our landscapes, but control for bias introduced by their different life-history, home-range estimates of each hornbill species in each habitat type were centred around the mean value and scaled to one-unit standard deviation. This was multiplied by a conversion factor of 465.3ha based on the mean home-range reported across the seven species (references above; Supplementary Data Table 1) to calculate scaled home range estimates for each of these species. Per-hectare density estimates were inferred as the inverse of scaled home-range.

Figure R2 Relationships between bird weight and territory size from a) average modeled estimates from this study, b) other published studies on tropical Asian birds c) hornbill studies in the literature.

- L 396: it wasn't clear to me what were the predicted variables in the model (guild energetics?), nor where the results were reported.

We have added text on this to the Methods (lines 436-440).

- L 413, 415, 421--423 – I didn't find any justification for the uncertainty values chosen in this section, e.g. why is mammals' body mass more variable (20%) than birds' (10%)? Where do all these uncertainty estimates come from?

In the initial manuscript these were educated guesstimates. For the birds, we now assign an uncertainty of 15% based on the standard error in the study by (Read et al. 2018) for tropical birds. We now apply the same 15% for mammals for consistency, in the absence of other data (lines 447-448).

Appropriate use of statistics and treatment of uncertainties

Some numbers reported in the main text (e.g. L 97, 152, 157) and in the figures lack uncertainty estimates. Is this to make the text more readable? If so, please provide this information elsewhere, or cite where it can be found (if in Supplementary Table 1 for example).

We have now added uncertainty estimates to some of these numbers, in particular around relative changes in energy flow between old growth and intact forests. We did not add uncertainty around every number presented, as for the less critical numbers they tended to disrupt readability and did not change the core arguments.

Error bars in Figures 1 and 3 should be described in the legends.

Error bar descriptions are now added in the figure legends.

L 118: The acronym CI should be spelled out and the range (95%?) reported.

Done.

L 195-197: the name of the test performed should be indicated, and the name of the statistic provided should be spelled out.

Done (lines 202-203).

Conclusions: robustness, validity, reliability

The conclusions seem to be supported by the data and by previous studies in the same study area. They could however be different in other regions with different species communities or under different disturbance regimes (e.g. with higher hunting pressure, as discussed in the main text).

We have added text discussing how this study represents a low-hunting scenario that represents the potential for amplification of ecosystem energetics if extensive hunting can be avoided (lines 218-229) and this point features in our conclusions (lines 269-272).

Suggested improvements: experiments, data for possible revision

See suggestions in section C for additional data. I don't think any additional experiments are needed.

Clarity and context: lucidity of abstract/summary, appropriateness of abstract, introduction and conclusions

The text is well-written, the abstract provides all the key information. The introduction provides essential context for this study, and the conclusions are concise and supported by the data analysis. I think, however, that the introduction could provide a little more context with other work on similar topics, in terms of the effect of land use/forest type on species diversity and abundance, and on ecosystem processes. For example, citing other studies that have examined the effect of different types and intensities of logging on wildlife diversity (e.g. Burivalova et al. 2014; Bicknell et al. 2014; full citations at end) or ecosystem functions (Schleuning et al. 2017).

We have included these references to improve context in the introduction, and also mention the Schleuning paper in our conclusions as supporting our findings.

Other minor comments

L18: missing parenthesis

Corrected

L 93 'biomass mass' is redundant

Corrected

L 114 – please provide a reference or data supporting this statement

A citation has been added (Kotowska et al. 2015)

L 410 – citation not in Nature style and not included in the References

This has been corrected.

Fig 1: (f) % NPP : y-axis not the right unit (should be % instead of kJ m⁻² year⁻¹)

Figure 1 – the colour legend doesn't correspond to the panel letters (a and d-g should respectively by e and c, d, f, g)

Figure 1 (now Figure 2) has been corrected and updated.

Figure S3: the legend should describe the colours used and their acronyms.

The Figure legend has been updated and clarified.

Figure S4: the panels mentioned in the legend don't correspond to the figure

The y-axis label was incorrect in the previous version and has now been corrected.

Almeida-Maués, Paula C. R., Anderson S. Bueno, Ana Filipa Palmeirim, Carlos A. Peres, and Ana Cristina Mendes-Oliveira. 2022. "Assessing Assemblage-Wide Mammal Responses to Different Types of Habitat Modification in Amazonian Forests." *Scientific Reports* 12 (1): 1797.

Asner, Gregory P., Philip G. Brodrick, Christopher Philipson, Nicolas R. Vaughn, Roberta E. Martin, David E. Knapp, Joseph Heckler, et al. 2018. "Mapped Aboveground Carbon Stocks to Advance Forest Conservation and Recovery in Malaysian Borneo." *Biological Conservation* 217 (January): 289–310.

Borchers, D. L., and M. G. Efford. 2008. "Spatially Explicit Maximum Likelihood Methods for Capture-Recapture Studies." *Biometrics* 64 (2): 377–85.

Brodie, Jedediah F., Anthony J. Giordano, and Laurentius Ambu. 2015. "Differential Responses of Large Mammals to Logging and Edge Effects." *Mammalian Biology = Zeitschrift Fur Säugetierkunde* 80 (1):

- Brown, James H., James F. Gillooly, Andrew P. Allen, Van M. Savage, and Geoffrey B. West. 2004. "TOWARD A METABOLIC THEORY OF ECOLOGY." *Ecology*. <https://doi.org/10.1890/03-9000>.
- Burivalova, Zuzana, Çağan Hakkı Sekercioğlu, and Lian Pin Koh. 2014. "Thresholds of Logging Intensity to Maintain Tropical Forest Biodiversity." *Current Biology: CB* 24 (16): 1893–98.
- Chapman, Philip M., Oliver R. Wearn, Terhi Riutta, Chris Carbone, J. Marcus Rowcliffe, Henry Bernard, and Robert M. Ewers. 2018. "Inter-Annual Dynamics and Persistence of Small Mammal Communities in a Selectively Logged Tropical Forest in Borneo." *Biodiversity and Conservation* 27 (12): 3155–69.
- Cleary, Daniel F. R., Timothy J. B. Boyle, Titiek Setyawati, Celina D. Anggraeni, E. Emiel Van Loon, and Steph B. J. Menken. 2007. "BIRD SPECIES AND TRAITS ASSOCIATED WITH LOGGED AND UNLOGGED FOREST IN BORNEO." *Ecological Applications*. <https://doi.org/10.1890/05-0878>.
- Croll, Donald A., Raphael Kudela, and Bernie R. Tershy. 2006. "Ecosystem Impact of the Decline of Large Whales in the North Pacific." *Whales, Whaling and Ocean Ecosystems*, 202–14.
- Edwards, David P., Paul Woodcock, Rob J. Newton, Felicity A. Edwards, David J. R. Andrews, Teegan D. S. Docherty, Simon L. Mitchell, et al. 2013. "Trophic Flexibility and the Persistence of Understory Birds in Intensively Logged Rainforest." *Conservation Biology: The Journal of the Society for Conservation Biology* 27 (5): 1079–86.
- Gale, George A., and Siriporn Thongaree. 2006. "Density Estimates of Nine Hornbill Species in a Lowland Forest Site in Southern Thailand." *Bird Conservation International* 16 (1): 57–69.
- Granados, Alys, Kyle Crowther, Jedediah F. Brodie, and Henry Bernard. 2016. "Persistence of Mammals in a Selectively Logged Forest in Malaysian Borneo." *Mammalian Biology = Zeitschrift Fur Säugetierkunde* 81 (3): 268–73.
- Hardwick, Stephen R., Ralf Toumi, Marion Pfeifer, Edgar C. Turner, Reuben Nilus, and Robert M. Ewers. 2015. "The Relationship between Leaf Area Index and Microclimate in Tropical Forest and Oil Palm Plantation: Forest Disturbance Drives Changes in Microclimate." *Agricultural and Forest Meteorology* 201 (February): 187–95.
- Kinnaird, Margaret F., and Timothy G. O'Brien. 2008. "Breeding Ecology of the Sulawesi Red-Knobbed Hornbill *Aceros Cassidix*." *Ibis*. <https://doi.org/10.1111/j.1474-919x.1999.tb04263.x>.
- Kotowska, Martyna M., Christoph Leuschner, Triadiati Triadiati, Selis Meriem, and Dietrich Hertel. 2015. "Quantifying Above- and Belowground Biomass Carbon Loss with Forest Conversion in Tropical Lowlands of Sumatra (Indonesia)." *Global Change Biology* 21 (10): 3620–34.
- L. Brooke, M. de. 2004. "The Food Consumption of the World's Seabirds." *Proceedings of the Royal Society of London. Series B: Biological Sciences* 271 (suppl_4): S246–48.
- Malhi, Yadvinder, Christopher E. Doughty, Gregory R. Goldsmith, Daniel B. Metcalfe, Cécile A. J. Girardin, Toby R. Marthews, Jhon Del Aguila-Pasquel, et al. 2015. "The Linkages between Photosynthesis, Productivity, Growth and Biomass in Lowland Amazonian Forests." *Global Change Biology* 21 (6): 2283–95.
- Malhi, Yadvinder, Cécile Girardin, Daniel B. Metcalfe, Christopher E. Doughty, Luiz E. O. C. Aragão, Sami W. Rifai, Immaculada Oliveras, et al. 2021. "The Global Ecosystems Monitoring Network: Monitoring Ecosystem Productivity and Carbon Cycling across the Tropics." *Biological Conservation* 253 (January): 108889.
- Milodowski, David T., David A. Coomes, Tom Swinfield, Tommaso Jucker, Terhi Riutta, Yadvinder Malhi, Martin Svátek, et al. 2021. "The Impact of Logging on Vertical Canopy Structure across a Gradient of Tropical Forest Degradation Intensity in Borneo." *The Journal of Applied Ecology*, no. 1365-2664.13895 (June). <https://doi.org/10.1111/1365-2664.13895>.
- Moore, Sam, Stephen Adu-Bredu, Akwasi Duah-Gyamfi, Shalom D. Addo-Danso, Forzia Ibrahim, Armel T. Mbou, Agnès de Grandcourt, et al. 2018. "Forest Biomass, Productivity and Carbon Cycling

- along a Rainfall Gradient in West Africa." *Global Change Biology* 24 (2): e496–510.
- Nyffeler, Martin, Çağan H. Şekercioğlu, and Christopher J. Whelan. 2018. "Insectivorous Birds Consume an Estimated 400–500 Million Tons of Prey Annually." *The Science of Nature* 105 (7): 47.
- O'Brien, Timothy G. 2008. "Behavioural Ecology of the North Sulawesi Tarictic Hornbill *Penelopides Exarhatus Exarhatus* during the Breeding Season." *The Ibis* 139 (1): 97–101.
- Poonswad, P., and A. Tsuji. 1994. "Ranges of Males of the Great Hornbill *Buceros Bicornis*, Brown Hornbill *Ptilolaemus Tickelli* and Wreathed Hornbill *Rhyticeros Undulatus* in Khao Yai National Park" *The Ibis*. https://onlinelibrary.wiley.com/doi/abs/10.1111/j.1474-919X.1994.tb08133.x?casa_token=UmoO4s_mjeYAAAAA:tFhPni2eLVuRfILuoG8QUhr2SU4foS6-WzxYtHRP4auiNyFIQfVHSazXLVfY6bho093ecMEmfr7zIOc.
- Portugal, Steven J., Jonathan A. Green, Lewis G. Halsey, Walter Arnold, Vincent Careau, Peter Dann, Peter B. Frappell, et al. 2016. "Associations between Resting, Activity, and Daily Metabolic Rate in Free-Living Endotherms: No Universal Rule in Birds and Mammals." *Physiological and Biochemical Zoology: PBZ* 89 (3): 251–61.
- Read, Quentin D., Benjamin Baiser, John M. Grady, Phoebe L. Zarnetske, Sydne Record, and Jonathan Belmaker. 2018. "Tropical Bird Species Have Less Variable Body Sizes." *Biology Letters* 14 (1). <https://doi.org/10.1098/rsbl.2017.0453>.
- Riutta, Terhi, Lip Khoo Kho, Yit Arn Teh, Robert Ewers, Noreen Majalap, and Yadvinder Malhi. 2021. "Major and Persistent Shifts in below-Ground Carbon Dynamics and Soil Respiration Following Logging in Tropical Forests." *Global Change Biology* 27 (10): 2225–40.
- Riutta, Terhi, Yadvinder Malhi, Lip Khoo Kho, Toby R. Marthews, Walter Huaraca Huasco, Minsheng Khoo, Sylvester Tan, et al. 2018. "Logging Disturbance Shifts Net Primary Productivity and Its Allocation in Bornean Tropical Forests." *Global Change Biology* 24 (7): 2913–28.
- Rowcliffe, J. Marcus, Juliet Field, Samuel T. Turvey, and Chris Carbone. 2008. "Estimating Animal Density Using Camera Traps without the Need for Individual Recognition." *The Journal of Applied Ecology* 45 (4): 1228–36.
- Royle, J. Andrew, J. Andrew Royle, and Kevin V. Young. 2008. "A HIERARCHICAL MODEL FOR SPATIAL CAPTURE–RECAPTURE DATA." *Ecology*. <https://doi.org/10.1890/07-0601.1>.
- Schoener, Thomas W. 1968. "Sizes of Feeding Territories among Birds." *Ecology* 49 (1): 123–41.
- Wearn, Oliver R., J. Marcus Rowcliffe, Chris Carbone, Marion Pfeifer, Henry Bernard, and Robert M. Ewers. 2017. "Mammalian Species Abundance across a Gradient of Tropical Land-Use Intensity: A Hierarchical Multi-Species Modelling Approach." *Biological Conservation* 212 (August): 162–71.

Reviewer Reports on the First Revision:

Referees' comments:

Referee #1 (Remarks to the Author):

The authors did a really great work in addressing my concerns in the previous version of the manuscript. The manuscript now (1) provides information on the distribution of sites across the gradient of aboveground biomass – clearly showing the low aboveground biomass in oil palm, followed by logged forests, and old-growth forests. It also (2) recognises the caveats regarding how defaunation may be higher in smaller and more isolated fragments of logged forests, and (3) shows LiDAR evidence that structure and biomass in heavily logged forests are different from some of their old-growth plots. Finally, they also (4) clarify that sampling effort differences across land-use classes are unlikely to affect their analysis outcomes, which are based on common species densities and account for imperfect detection by including camera trap nights in the REM models. Therefore, I am confident that their key message on the higher abundances and energetics of logged forests compared to old-growth forests is supported by their data and analysis. I also appreciate the authors recognise the ecological importance of logged forests (e.g. compared with palm oil or pastures), from an energetic perspective, is highlighted without diminishing the importance of undisturbed tropical forests to safeguard global biodiversity. Therefore, I have no further requests and believe this manuscript would make a great contribution to Nature.

Dr Filipe França

Referee #2 (Remarks to the Author):

I very much appreciate the effort and quality of the responses to reviewer comments and the additions to the manuscript. My main comments (I was Rev #2 in the first round) on dietary shifts and representativeness were treated well. I particularly value the new hunting scenarios, and am convinced about the robustness of the results based on these additional analyses.

I'd still be in favour of moving the oil palm results to Suppl Materials, because these do not contribute to the title and do not make it to the abstract, but I rest my case.

Some minor comments:

L 202-203: "of 50% reduction of species" is unclear; I think meant is to indicate a 50% biomass reduction, but this phrase could also suggest reducing the # of species by 50%.

L424-426: I suggest adding a reference on possible dietary shifts in response to logging, to support the 30% change scenario.

L424-426: I could not find the results of the 30% change in feeding scenario. The line numbers in the response letter (L709 in Suppl Mat) do not exist.

L465-466: perhaps add to this final sentence what this implies in terms of energetics (a smaller energetics increase due to logging?)

Referee #3 (Remarks to the Author):

The authors have submitted a substantially revised version of their manuscript. In particular, the change of the first figure and the extended description of the methods have improved the manuscript. However, I still do not think that the points raised in my prior review have been fully addressed. I have tried to explain them more explicitly in this review.

Spatial design of the study and spatial auto-correlation

1) I raised two points in my previous review, regarding the spatial arrangement of the study and its implications. The authors have responded by (1) adding a figure (revised Fig. 1) showing the spatial location of the study plots, and (2) comparing the impacts in the four old-growth forests.

First, I am quite pleased with Figure 1, which now allows us to assess the spatial design of the study. This also helps to answer my point related to the NPP measurements. However, I must note that the presentation is not of the quality of a publication (e.g., the legend is not adequate and the labeling of the panels is not complete). The figure legend needs to be improved to include all the information needed to understand the figure.

Second, I do not think the author's response to my comment on spatial auto-correlation is adequate. As mentioned in my original comment, the logged sites are much closer together (now visible in Fig. 1) than the old-growth forests ("I get the impression that only one logged forest was studied"). In response to this comment, it is not sufficient to compare the old forests. Instead, I propose to use a spatial auto-correlation function to account for the greater proximity of the logged plots to each other. Without such a statistical procedure, which is quite common in this field, the conclusions may not be valid.

Description of statistical approaches

2) In response to my previous comment, the authors have added a brief description of their statistical approaches to the manuscript (lines 400 - 404). I am sorry to be picky here, but the information given here is not sufficient to replicate the study. What kind of random effect was used? How exactly have the unequal variances (how unequal) been accounted for? I think this is quite a minor point that can easily be fixed but it requires attention.

Calculation of consumption rates

3) In my previous review, I raised concerns about the calculation of energy fluxes related to the effects of (1) temperature and (2) species interactions.

First, the authors provide an excellent argument in their response as to why it is not essential to consider temperature variations. I think this should be more clearly highlighted in the manuscript rather than hidden in the review response.

Second, there is no response to my comment that interactions are important for calculating energy fluxes. The latter point is based on the pattern that populations can be driven to low densities by predation despite high production rates (and thus high consumption rates and energy fluxes). A model like the one used in this study ignores this possibility and is therefore too simplistic." I realize this is a difficult point, but without accounting for these energy flows to consumers of a species, the calculation represents short-term steady-state consumption rates. I think this differentiation is important (also to allow comparisons between studies) and should be discussed in the main manuscript.

Third, I must add that the description in the methods is too brief. The allometric equations are taken from Nagy et al. (1999, cited in the manuscript). It should be mentioned that these equations estimate field metabolic rates, and it needs to be mentioned in the methods which equations and which parameters were used for this calculation. In addition, there is no description of how "the proportions of each food type in the diet were assigned to each species" (lines 389-390) and the source of the assimilation rates (line 390). All of these details are important to repeat the study.

Support of the main conclusions by energy consumption rates of birds and mammals

4) In my previous review, I expressed concern that the general statements in the conclusions and abstract were not fully supported by the analysis of mammalian and avian consumption rates. In my opinion, the authors did a good job addressing this comment in most parts of the manuscript. However, I do not think that the argument regarding increases in insectivorous birds from old-growth forests to logged forests is very convincing. For example, many insects (especially herbivorous ones) in tropical forests are tightly controlled by their parasitoid enemies. Deforestation creates more open forest structures that often favor birds and mammals and thus increases their density. Based on the data collected, it cannot be ruled out that energy flows shift from those within the insect community (e.g., from herbivorous insects to parasitoids) to those from insects to insectivorous birds. I think such a critical evaluation of the data collected and its constraints on the take-home message should be part of the discussion.

Following up on this, I would question the title of the manuscript. I see support that the analyses show changes or diversification in bird and mammal consumption rates (which needs to be defined at some place in the introduction or abstract as the measurement of ecosystem energetics) but I see less support for an amplification if ecosystem energetics is seen as anything beyond the diets of birds and mammals. I have the impression that most readers will interpret this title to expect something different compared to the content of the manuscript.

Referee #4 (Remarks to the Author):

The authors have addressed my main concerns as well as those of other reviewers, with reasonable answers and additional analyses that make their results more robust. They did a good job of clarifying the manuscript, especially in the methods, and the new Figure 1 helps to provide a better overview of the extensive sampling effort in the study area.

Figure 1 also made me aware of the distance between OG NPP plots and logged NPP plots, which questions the comparability of NPP values between these two land uses in the study. The authors mentioned in their response to the reviewers that bird and mammal energetics show consistent differences between logged and OG forests within the SAFE project area, but since there is no OG NPP plot on SAFE project area (the only logged site), there is no way to know if this is true for NPP. This should not fundamentally change the results as NPP is not the main focus of this study, but it could be worth mentioning in the manuscript (perhaps in the Caveats section).

Another minor comment on Figure 1: oil palm plantations are not mentioned in the legend, I guess it is the white area (by default) but it should be explicitly added.

Overall, the manuscript seems much stronger and I recommend accepting it after minor revisions.

Author Rebuttals to First Revision:

Response to Referee Comments

We thank all referees for the new comments, and are very pleased that the recommend acceptance. We respond to the remaining comments in **green italics** below.

Referees' comments:

Referee #1 (Remarks to the Author):

The authors did a really great work in addressing my concerns in the previous version of the manuscript. The manuscript now (1) provides information on the distribution of sites across the gradient of aboveground biomass – clearly showing the low aboveground biomass in oil palm, followed by logged forests, and old-growth forests. It also (2) recognises the caveats regarding how defaunation may be higher in smaller and more isolated fragments of logged forests, and (3) shows LiDAR evidence that structure and biomass in heavily logged forests are different from some of their old-growth plots. Finally, they also (4) clarify that sampling effort differences across land-use classes are unlikely to affect their analysis outcomes, which are based on common species densities and account for imperfect detection by including camera trap nights in the REM models. Therefore, I am confident that their key message on the higher abundances and energetics of logged forests compared to old-growth forests is supported by their data and analysis. I also appreciate the authors recognise the ecological importance of logged forests (e.g. compared with palm oil or pastures), from an energetic perspective, is highlighted without diminishing the importance of undisturbed tropical forests to safeguard global biodiversity. Therefore, I have no further requests and believe this manuscript would make a great contribution to Nature.

Dr Filipe França

We thank Reviewer 1 for the time and thought they put into their review and are pleased he finds it a great potential contribution to Nature. His comments greatly strengthened the paper.

Referee #2 (Remarks to the Author):

I very much appreciate the effort and quality of the responses to reviewer comments and the additions to the manuscript. My main comments (I was Rev #2 in the first round) on dietary shifts and representativeness were treated well. I particularly value the new hunting scenarios, and am convinced about the robustness of the results based on these additional analyses.

We thank Reviewer 2 for their questions and suggestions, which we believe greatly improved our analysis and the strength of the paper.

I'd still be in favour of moving the oil palm results to Suppl Materials, because these

do not contribute to the title and do not make it to the abstract, but I rest my case.

Some minor comments:

L 202-203: "of 50% reduction of species" is unclear; I think meant is to indicate a 50% biomass reduction, but this phrase could also suggest reducing the # of species by 50%.

We meant 50% reduction in population density of those species sensitive to targeted and/or indiscriminate hunting (this is the same as 50% reduction of biomass of those species), and have clarified this in the manuscript (lines 215-217).

L424-426: I suggest adding a reference on possible dietary shifts in response to logging, to support the 30% change scenario.

We performed the 30% diet shift analysis as a sensitivity test and conclude that diet shifts would have little overall effect on the results. We could find very little solid evidence in the literature of shifts in food type in response to logging. The Edwards et al. (2013) paper (Reference #18) shows within-species changes towards higher trophic levels in response to logging for some birds, and Magioli et al. (2019) found evidence of a diet shift in some mammals, but that was in the context of a surrounding agricultural matrix that became an alternative food source.

L424-426: I could not find the results of the 30% change in feeding scenario. The line numbers in the response letter (L709 in Suppl Mat) do not exist.

This analysis had been accidentally. It showed a negligible shift in direct consumption but a slightly larger potential shift in indirect consumption. The following text has now been added to the supplementary material (lines 71-82).

“The 30% shift towards invertebrate consumption at the expense of other food groups in logged forest results in a negligible 0.20% increase in total energy consumption. This is due to the energetic properties of all food types being very similar, except leaves. and very few species in our dataset feeding on both leaves and invertebrates. However, this diet shift would lead to a bigger change, 8.9%, in the amount of NPP required to support insectivory (assuming a mean trophic level of consumed invertebrates to be 2.5), from 51% of NPP to 55% of NPP. “

L465-466: perhaps add to this final sentence what this implies in terms of energetics (a smaller energetics increase due to logging?)

We have added the following to the final sentence of Methods " and would therefore show a smaller increase in energetics." (lines 547-548).

Referee #3 (Remarks to the Author):

The authors have submitted a substantially revised version of their manuscript. In particular, the change of the first figure and the extended description of the methods have improved the manuscript. However, I still do not think that the points raised in my prior review have been fully addressed. I have tried to explain them more explicitly in this review.

We thank the reviewer for taking the time and thought to further clarify their points, which we address below.

Spatial design of the study and spatial auto-correlation

1) I raised two points in my previous review, regarding the spatial arrangement of the study and its implications. The authors have responded by (1) adding a figure (revised Fig. 1) showing the spatial location of the study plots, and (2) comparing the impacts in the four old-growth forests.

First, I am quite pleased with Figure 1, which now allows us to assess the spatial design of the study. This also helps to answer my point related to the NPP measurements. However, I must note that the presentation is not of the quality of a publication (e.g., the legend is not adequate and the labeling of the panels is not complete). The figure legend needs to be improved to include all the information needed to understand the figure.

We have improved the figure labelling and legend to provide a fuller description of the information needed to understand the figure.

Second, I do not think the author's response to my comment on spatial auto-

correlation is adequate. As mentioned in my original comment, the logged sites are much closer together (now visible in Fig. 1) than the old-growth forests ("I get the impression that only one logged forest was studied"). In response to this comment, it is not sufficient to compare the old forests. Instead, I propose to use a spatial autocorrelation function to account for the greater proximity of the logged plots to each other. Without such a statistical procedure, which is quite common in this field, the conclusions may not be valid.

We have considered and discussed this suggestion among co-authors and do not believe that an appropriate spatial autocorrelation analysis is possible or would yield anything meaningful. In our study we have aggregated species abundance data to be one data point per habitat type. We do this aggregation because we are combining data across taxa for which we need the largest sampling effort and 'best' description of the community possible. Hence our unit of replication is guild, which has no spatial component to it. We do not present any statistical analyses which have site as the unit of replication, which is the only context in which spatial autocorrelation can influence p-values.

Working at individual sampling point scale would necessarily mean every single site is under-sampled, so we would not gain anything particularly reliable from the analysis. Hence, we do not have spatial replicates within habitat type so we do not have a spatial autocorrelation variable that can be explored. The closest we have is the replicates of old-growth forest, which we have already presented.

Description of statistical approaches

2) In response to my previous comment, the authors have added a brief description of their statistical approaches to the manuscript (lines 400 - 404). I am sorry to be picky here, but the information given here is not sufficient to replicate the study. What kind of random effect was used? How exactly have the unequal variances (how unequal) been accounted for? I think this is quite a minor point that can easily be fixed but it requires attention.

We have now added this additional detail into the main text (lines 471-476).

Calculation of consumption rates

3) In my previous review, I raised concerns about the calculation of energy fluxes related to the effects of (1) temperature and (2) species interactions.

First, the authors provide an excellent argument in their response as to why it is not essential to consider temperature variations. I think this should be more clearly highlighted in the manuscript rather than hidden in the review response.

We have now added a section on temperature variations in the supplementary material (lines 106-129), replicating the points that were made in our previous response to reviewers.

Second, there is no response to my comment that interactions are important for calculating energy fluxes. The latter point is based on the pattern that populations can be driven to low densities by predation despite high production rates (and thus high consumption rates and energy fluxes). A model like the one used in this study ignores this possibility and is therefore too simplistic." I realize this is a difficult point,

but without accounting for these energy flows to consumers of a species, the calculation represents short-term steady-state consumption rates. I think this differentiation is important (also to allow comparisons between studies) and should be discussed in the main manuscript.

*As pointed out by the referee, **short-term steady state consumption rates** are exactly what we mean by ecological energetics in this study, i.e. what is being consumed by the populations of animal and bird species at any one time period. This is a measure of the energy flow across different guilds and functional groups. We have now clarified this definition at the start of the manuscript to prevent any confusion or ambiguity (lines 71-73):*

“We interpret ecological energetics to be the short-term equilibrium production or consumption rates of food energy by specific species, guilds or taxonomic groups.”

Third, I must add that the description in the methods is too brief. The allometric equations are taken from Nagy et al. (1999, cited in the manuscript). It should be mentioned that these equations estimate field metabolic rates, and it needs to be mentioned in the methods which equations and which parameters were used for this calculation. In addition, there is no description of how "the proportions of each food type in the diet were assigned to each species" (lines 389-390) and the source of the assimilation rates (line 390). All of these details are important to repeat the study.

The text has now been expanded. The equations and parameters have been added to the Supplementary Data (Table S1), and referred to in the main text. The assimilation efficiencies were already listed in the Supplementary Data file but are

now also given in the Supplementary Material (Table S2). Diet proportions were based on expert knowledge (with appropriate uncertainty estimates) of species-specific feeding ecology by the co-authors (particularly OW, NJD and SM). This detail has now been added to Methods (lines 456-459).

Support of the main conclusions by energy consumption rates of birds and mammals

4) In my previous review, I expressed concern that the general statements in the conclusions and abstract were not fully supported by the analysis of mammalian and avian consumption rates. In my opinion, the authors did a good job addressing this comment in most parts of the manuscript. However, I do not think that the argument regarding increases in insectivorous birds from old-growth forests to logged forests is very convincing. For example, many insects (especially herbivorous ones) in tropical forests are tightly controlled by their parasitoid enemies. Deforestation creates more open forest structures that often favor birds and mammals and thus increases their density. Based on the data collected, it cannot be ruled out that energy flows shift from those within the insect community (e.g., from herbivorous insects to parasitoids) to those from insects to insectivorous birds. I think such a critical evaluation of the data collected and its constraints on the take-home message should be part of the discussion.

The reviewer is suggesting that there is a large energy flow from herbivorous insects to parasitoids in old-growth forests, which is intercepted or disrupted by birds and mammals in logged forests. We think the suggestion that there is a massive parasitoids collapse (the collapse and diversion would need to be massive to support the changes we observe in vertebrate insectivory) and diversion of this energy to

vertebrates seems unlikely, and there is very little evidence to support this conjecture. Certainly there will be a portion of energy that does not end up consumed by vertebrates, but there is no evidence or reason to expect that this proportion will change dramatically between primary and logged forest. We have added a sentence to the main text raising the reviewer's suggestion as a possibility that could be explored further (lines 243-246) although it is unlikely to be an explanation for our observations. As this mechanism seems very unlikely we have not further elaborated on this point in the limited space available in the discussion. However, we provide more detail below to support our position and some of these points could be made in supplementary material if deemed useful.

.

The very limited literature we were able to find suggests there may be a decline of parasitoids in logged forest (e.g. (Hopkins, Roininen, and Sääksjärvi 2019) for one group, Rhyssinae, in an African rainforest). We think that any observed decline in parasitoids could happen for four possible reasons, but none of them provide strong support for the idea that energy would be significantly diverted from invertebrates to vertebrates when moving from old growth to logged forest:

- 1) Increased consumption of herbivorous insects by vertebrates in the logged forest, before parasitoids can develop. This seems unlikely because many parasitoids develop with host larvae at a very early stage.*
- 2) An environmental factor (e.g. humidity or light) that causes parasitoid populations to decline substantially in logged forests, so that more herbivores are*

available to vertebrates. We know of no evidence of this, and it seems implausible that parasitoids are more sensitive to environmental factors than other invertebrates.

3) Heavy predation of parasitoids themselves by vertebrates in the logged forest.

This would increase the calculated volume of herbivorous insects required to support the parasitoid population that is being consumed and would require an even larger estimated population of herbivorous insects in the logged forest. This would result in larger indirect energetic flows through the vertebrate channels we examine in logged relative to old growth forest, which is the opposite to what the reviewer suggests.

Such a mechanism would not support the hypothesis of diversion of energy from parasitoids to vertebrates. The possible increased predation of parasitoids is further supported by evidence that the trophic level of insectivorous bird diets in our study region increases substantially in logged forests (Edwards et al. 2013).

4) A decline in host populations of herbivorous insects, which is inconsistent with our observation of greatly increased insectivory. Such a mechanism would not support the hypothesis of diversion of energy from parasitoids to vertebrates.

Following up on this, I would question the title of the manuscript. I see support that the analyses show changes or diversification in bird and mammal consumption rates (which needs to be defined at some place in the introduction or abstract as the measurement of ecosystem energetics) but I see less support for an amplification if ecosystem energetics is seen as anything beyond the diets of birds and mammals. I have the impression that most readers will interpret this title to expect something different compared to the content of the manuscript.

We have now clarified our definition of ecosystem energetics (see above). However, we feel the explanation we present in the manuscript (increased palatability of plant material driving increased insect abundances) seems by far the most parsimonious and likely explanation for the observed phenomenon (see our response to the parasitoid comment above) , and is also supported by the increased insect abundances reported at logged forests in this study system (Ewers et al. 2015). Hence, we are reluctant to change our title as we believe it captures our main point well without inserting “bird and mammal consumption energetics” (which is clunky would and also not comply with the 75 character-limit for the title – the current title is 63 characters).

Referee #4 (Remarks to the Author):

The authors have addressed my main concerns as well as those of other reviewers, with reasonable answers and additional analyses that make their results more robust. They did a good job of clarifying the manuscript, especially in the methods, and the new Figure 1 helps to provide a better overview of the extensive sampling effort in the study area.

We thank the reviewer for their thoughtful comments and contributions that have improved the manuscript.

Figure 1 also made me aware of the distance between OG NPP plots and logged NPP plots, which questions the comparability of NPP values between these two land uses in the study. The authors mentioned in their response to the reviewers that bird

and mammal energetics show consistent differences between logged and OG forests within the SAFE project area, but since there is no OG NPP plot on SAFE project area (the only logged site), there is no way to know if this is true for NPP. This should not fundamentally change the results as NPP is not the main focus of this study, but it could be worth mentioning in the manuscript (perhaps in the Caveats section).

We now mention this point in the caveats section (lines 532-536), but also state that the variation of measured NPP across old growth sites in NE Borneo with very strongly varying soil conditions (from infertile sandy soils to through clay oxisols to forests of basic basalt outcrops) is fairly small (range 12.03 – 15.53 Mg C ha⁻¹ year⁻¹ (so. ±9%) Riutta et al 2018). We therefore fully expect that NPP in old-growth forests in the SAFE landscape will be within this range. The SAFE landscape is located between the Maliau and Danum old-growth sites and is unlikely to be an outlier in terms of geology or productivity.

Another minor comment on Figure 1: oil palm plantations are not mentioned in the legend, I guess it is the white area (by default) but it should be explicitly added.

We have now mentioned oil palm as white in the figure legend.

Overall, the manuscript seems much stronger and I recommend accepting it after minor revisions.

- Persistence of Understory Birds in Intensively Logged Rainforest.” *Conservation Biology: The Journal of the Society for Conservation Biology* 27 (5): 1079–86.
- Ewers, Robert M., Michael J. W. Boyle, Rosalind A. Gleave, Nichola S. Plowman, Suzan Benedick, Henry Bernard, Tom R. Bishop, et al. 2015. “Logging Cuts the Functional Importance of Invertebrates in Tropical Rainforest.” *Nature Communications* 6 (April): 6836.
- Hopkins, Tapani, Heikki Roininen, and Ilari E. Sääksjärvi. 2019. “Extensive Sampling Reveals the Phenology and Habitat Use of Afrotropical Parasitoid Wasps (Hymenoptera: Ichneumonidae: Rhyssinae).” *Royal Society Open Science* 6 (8): 190913.
- Magioli, Marcelo, Marcelo Zacharias Moreira, Renata Cristina Batista Fonseca, Milton Cezar Ribeiro, Márcia Gonçalves Rodrigues, and Katia Maria Paschoaletto Micchi de Barros Ferraz. 2019. “Human-Modified Landscapes Alter Mammal Resource and Habitat Use and Trophic Structure.” *Proceedings of the National Academy of Sciences of the United States of America* 116 (37): 18466–72.

Reviewer Reports on the Second Revision:

Referees' comments:

Referee #3 (Remarks to the Author):

I think this is a very interesting manuscript that addresses an important question. Because the results could spark a very controversial debate, I reviewed the manuscript to help make the results bulletproof. Reading the authors' responses to my comments, I get the impression that we are going around in circles and that some of my comments were seen as rather annoying. I am sorry if my comments seemed too critical and focused on details. If anything, I don't want to be the obstacle that has to be overcome, because that's probably not the most productive way to advance our science. I do not want to hinder the further process and have only added a few minor points of clarification below.

General presentation

Most of my problems arise as I am trying to follow the methods and I am sorry to report this that this this keeps being a source of frustration. I have the impression that this manuscript was prepared hastily and without due attention to detail. The addition of a statistical section on the methods and the description of the spatial location of the study sites have helped, but I still cannot fully comprehend the methods as described in my point below. As an aside, the line numbers in the response to my review do not match the text in the manuscript that is referenced, which gives an idea of why I think the manuscript should be more thoroughly prepared before resubmitting it.

Spatial design of the study and spatial autocorrelation

In my two prior reviews, I have raised a point related to the spatial arrangement of the study and its implications. This point is related to a concern that the study should be (a) repeatable and (b) test its hypothesis. The first part has been responded to very well by adding a statistical methods section and the figure showing the location of the sites. As for the second part, I have been concerned that it should be shown that similarities between old-growth and between logged sites are driven by the difference in the land use (as phrased in the hypothesis) and not by the fact that the old-growth forests are in close spatial proximity to each other. You have responded that "we have aggregated species abundance data to be one data point per habitat type." This leads to the conclusion that "We do not present any statistical analyses which have site as the unit of replication". I have to admit that based on my reading I have to respond that this point is not evident at all from the description of the methods. I do not want to be the picky reviewer that keeps riding a wave and give up at this point but as a gentle suggestion, it might be worth checking the methods section that it really describes what has been done. For instance, there are sentences such as "A buffer of 100 m around the trap locations defined the region of model integration" (lines 385-386), and "Therefore, densities were derived based on a 10m detection radius (i.e., 0.126 ha) around each trap,...". This is giving me the impression that the camera sites have been used for estimating densities in a spatially-explicit way. To avoid an extra loop, I suggest adding a short introduction to the methods stating what is being measured (in the field) and how the data is aggregated.

Author Rebuttals to Second Revision:

Response to Reviewer 3

We are certainly very grateful for all the care and attention that the Reviewer has given to our work and appreciate their intention to make this work as robust and clearly explained as possible. Their comments and inputs have undoubtedly improved the clarity of our manuscript and will greatly help in response to any future critiques and discussions.

We realise we were not as clear in our description of data aggregation approaches as we could have been. As requested by the Reviewer, we have now added a short section in Methods that explains this explicitly (as well as tweaking other lines in Methods to further clarify this). We have also carefully gone through the Methods and main text to make sure it is as clear as possible.

In lines 445-457 we have added the following new section to Methods

Aggregation of species density data to habitat type

For all taxa, we estimated a single density value for each habitat type (old-growth forest, logged forest and oil palm), using all available data to do so. The 'aggregation' method used to achieve this differed by taxon. For small mammals, we used all available data for each habitat type to fit a SECR model (three models in total for each species), outputting density estimates (with uncertainty) for each habitat. For bats, a total sampling area was calculated (based on 20m buffers around each trap) for each habitat type and the total number of bats of each species was divided by this area in order to obtain a density estimate for each habitat type. For birds and medium/large terrestrial mammals, a completely model-based approach was used, i.e. multi-species models with categorical variables for habitat type. These models yielded habitat-specific density estimates (with uncertainty) for each species in the model. Our unit of replication in this analysis is therefore guild, which has no spatial component within habitat type and hence no spatial autocorrelation variable that can be explored.

Other key new lines include lines 373-375

For the terrestrial medium/large mammals, we estimated density using the Random Encounter Model (REM), with a categorical covariate included for habitat type (old-growth forest, logged forest and oil palm).

And lines 419-420:

This approach was directly analogous to the approach used in the REM density modelling (described above).

Reviewer Reports on the Third Revision:

Referees' comments:

Referee #3 (Remarks to the Author):

This has been a long process of comments and revisions. On a personal note, I can say that I am sorry that I caused the authors a lot of trouble by my constant requests for clarification of the methods.

I can now say that the methods provide the details necessary to follow the study design, and the results show a clear story. I must admit that there are a few points where I would follow a different policy, including the sample design and the statistical approach to spatial autocorrelation. However, I do not believe that an extended discussion during the review process is a fruitful avenue, as we would keep repeating our opinions and there is no general standard on how to approach these points. Despite our discrepancies, I think this study is important.

Therefore, I do not have any further points to make and congratulate the authors on this fine study and wish for interesting follow-ups.